# Nanoparticles for Topical Application in the Treatment of Skin Dysfunctions—An Overview of Dermo-Cosmetic and Dermatological Products

**DOI:** 10.3390/ijms232415980

**Published:** 2022-12-15

**Authors:** Magdalena Raszewska-Famielec, Jolanta Flieger

**Affiliations:** 1Faculty of Physical Education and Health, University of Physical Education, Akademicka 2, 21-500 Biała Podlaska, Poland; 2Department of Analytical Chemistry, Medical University of Lublin, Chodźki 4A, 20-093 Lublin, Poland

**Keywords:** nanotechnology, nanoparticles, nanomaterials, skin disorders, dermatology, cosmetology, drug delivery systems, nanocosmeceuticals, nanopharmaceuticals

## Abstract

Nanomaterials (NM) arouse interest in various fields of science and industry due to their composition-tunable properties and the ease of modification. They appear currently as components of many consumer products such as sunscreen, dressings, sports clothes, surface-cleaning agents, computer devices, paints, as well as pharmaceutical and cosmetics formulations. The use of NPs in products for topical applications improves the permeation/penetration of the bioactive compounds into deeper layers of the skin, providing a depot effect with sustained drug release and specific cellular and subcellular targeting. Nanocarriers provide advances in dermatology and systemic treatments. Examples are a non-invasive method of vaccination, advanced diagnostic techniques, and transdermal drug delivery. The mechanism of action of NPs, efficiency of skin penetration, and potential threat to human health are still open and not fully explained. This review gives a brief outline of the latest nanotechnology achievements in products used in topical applications to prevent and treat skin diseases. We highlighted aspects such as the penetration of NPs through the skin (influence of physical–chemical properties of NPs, the experimental models for skin penetration, methods applied to improve the penetration of NPs through the skin, and methods applied to investigate the skin penetration by NPs). The review summarizes various therapies using NPs to diagnose and treat skin diseases (melanoma, acne, alopecia, vitiligo, psoriasis) and anti-aging and UV-protectant nano-cosmetics.

## 1. Introduction

Nanoparticles (NPs) are defined as materials with dimensions smaller than 100 nm and presenting various shapes, i.e., spheres, rods, dendritic shapes, etc. [1]. This definition is accepted by the European Union (EU) Commission [2]. It should be noted, however, that there exists no uniform definition of nanomaterials [3]. The Environmental Protection Agency (EPA) emphasizes, in its opinion, the unique properties of NPs, which largely differentiate them from equivalent chemical compounds [4]. In turn, the US Food and Drug Administration (USFDA) clearly states that NPs should exhibit dimension-dependent phenomena [5]. The International Organization for Standardization (ISO), as the basic criterion, considers the nanoscale dimension of both the external dimension as well as the internal surface structure [6].

Naturally occurring nanostructures include allergens [7], microorganisms, i.e., viruses and bacteria [8,9], but also NPs formed during volcanic eruptions [10]. In the human body, there are numerous nanostructures without which the normal functioning of the body is impossible, i.e., enzymes, proteins, antibodies, or DNA. Human bone, which is a multifaceted composite of hierarchical inorganic nanohydroxyapatite and organic collagen, can also be classified as a nanomaterial [11,12]. In the anthropogenic environment, one can find atmospheric NPs produced as a result of industrial activity, i.e., exhaust fumes, smoke, and dust [13,14].

The history of synthetic NMs begins 4500 years ago in ancient Egypt [15]. Probably one of the first synthetic NMs was lead(II) sulfide NPs (5 nm) (PbS-NPs) used for dyeing or the so-called “Egyptian blue”, being a mixture of cuprorivaite CaCuSi_4_O_10_ and silicon dioxide (SiO_2_). The first scientific report describing the synthesis of gold NPs (Au-NPs) was made by Michael Faraday in 1857.

Generally, NMs are classified into four categories: carbon nanomaterials, inorganic, organic, and composite-based nanomaterials. Technologically produced nanotubes, fullerenes, quantum dots (QD), metals (silver Ag, gold Au), metal oxides (titanium dioxide TiO_2_, zinc oxide ZnO, iron (III) oxide Fe_2_O_3_, SiO_2_), and lipophilic NPs find more and more applications in cosmetics. This is due to the fact that NPs, thanks to their high surface-to-volume ratio [16], in addition to interesting physicochemical, electronic, optical, mechanical, catalytic, and thermal properties, also help in better penetration through the skin barrier [17].

Nanoparticles are ubiquitous in cosmetic products as antioxidants and anti-reflectants. Examples include TiO_2_-NPs added to creams as a white pigment or Ag-NPs as a component of shampoos and toothpaste [3]. In 1986, Christian Dior developed the first lysosomal anti-aging cream—Capture [18].

Many applications of nanoparticles have been described, not only in cosmetics but also in preparations for the treatment of skin diseases [19]. In nanomedicine, liposomal systems for transdermal drug delivery [20,21], contrast agents for diagnosing diseases, and gene therapies for cancer treatment have gained popularity [22,23,24,25,26].

An example of the use of NPs in medicine is Fe_2_O_3_-NPs used as a contrast in magnetic resonance imaging (MRI) [22]. Fe_2_O_3_-NPs, similarly to other magnetic nanoparticles (MNPs), besides their use as MRI contrast agents, can be used as vehicles, combined with superconductors, in magnetic drug delivery systems (MDDS). Due to the possibility of precision-guiding MNPs by an external magnetic field to the required area, MDDS has become promising in cancer therapy. MNPs can not only effectively transport and deliver drugs with a high concentration in cancerous tissues, but also generate heat through the oscillation of their magnetic pulse (44–47 °C), enabling the process of thermoablation of cancer cells (magnetic hyperthermia) [23].

In view of the growing trend of applying NMs in medicine, there is also an intensified interest in their toxic side effects, especially of those NPs that are not biodegradable, i.e., NPs of metals and metal oxides (in contrast to biodegradable NPs prepared from a variety of materials such as lipids, proteins, polysaccharides, and synthetic biodegradable polymers such as starches, chitin/chitosan, or poly-(D,L-lactide-coglycolide). Obtaining a therapeutic effect in the dermal or transdermal administration of drugs or cosmetic preparations chiefly depends on passing through the skin barrier [27]. NMs in biomedical applications are characterized by high bioactivity and bioavailability. Unfortunately, such features may prove to be a threat in the event of potential toxicity. Previous studies have shown that exposure to NPs contributes to the generation of reactive oxygen species (ROS) [28], as well as cytotoxicity and genotoxicity [29,30,31]. In vitro studies have shown that the cytotoxic effects of NPs may derive from many factors such as chemistry, dose and exposure time, particle size particle shape, aggregation, surface area, crystal structure, surface functionalization, and pre-exposure effects [29,32,33,34], which are crucial for optimizing potential applications. It should not be forgotten that the availability of pharmaceuticals, as regards topical administration, is rather limited to the organelles of the skin, i.e., hair follicles, sweat glands, and sebaceous glands. In this case, the systemic circulation is bypassed, which reduces adverse or toxic reactions.

It turns out, nevertheless, that the benefits of using NPs outweigh potential concerns related to the toxicity of NMs. The review article written by Gupta et al. [35] summarizes the regulatory guidelines and recommendations concerning the safe use of nanocosmeceutical products in India, Europe, and the USA. Since 2006, the FDA [36] and 2013, the EU [37] have been collecting data on the impact of NPs on humans and the environment. An example of such a study is the work of Lee et al. [38] describing the relationship between markers of oxidative stress, i.e., urinary 8-hydroxy-2′-deoxyguanosin (8-OHdG) concentration and the creatinine-adjusted concentration, and the exposure of cosmetics and clothing sellers potentially exposed to TiO_2_-NPs and ZnO-NPs. It appeared that the co-exposure index was significantly positively associated with both markers, reflected by β = 0.308, 95% CI from 0.106 to 0.510, and β = 0.486, 95% CI from 0.017 to 0.954, respectively. Furthermore, participants with exposure to NPs had a statistically higher level of 8-OHdG in urine in comparison to the lower co-exposure group (5.82 vs. 2.85 ng/mL, *p* < 0.001). Other studies [39] also confirmed higher levels of 8-OHdG and inflammatory markers such as cytokines IL-6, IL-8, and TNF-α after exposure to NPs. There is no complete agreement on the penetration of TiO_2_-NPs and ZnO-NPs through the epidermis into the bloodstream or whether this causes long-term toxicity [40].

This review discusses various NPs used in dermatological and cosmetic products, highlighting their application in the therapy of different skin diseases (acne, psoriasis, vitiligo, alopecia, skin cancer), as well as in cosmetology as anti-aging and UV-protecting agents. The review focuses on NMs intended for topical application to the skin for protective and healing purposes. It discusses the penetration of NPs through the skin and the parameters influencing this process. Furthermore, the present review highlights the experimental models used for in vivo and in vitro studies of skin penetration and the potential threats to the environment associated with the production of NMs. Literature was searched in PubMed, Scopus, Google Scholar, and Web of Science databases using key search terms, i.e., nanotechnology, nanoparticles, nanomaterials, skin disorders, dermatology, cosmetology, drug delivery systems, nanocosmeceuticals, nanopharmaceuticals. 

## 2. The Penetration of NPs through the Skin

### 2.1. The Routes of Penetration

The skin consists of several heterogeneous layers, i.e., the epidermis, the dermis, the hypodermis, and its appendages (hair follicles, sweat glands, sebaceous glands). The epidermis consists of keratinocytes stratified from a basal layer of viable cells to an outermost layer of terminally differentiated keratinocytes. The stratum corneum (SC) is a very thin layer of about 10 µm and comprises the three main components: (1) natural-moisturizing-factor (NMF)-laden and lipid-bound corneocytes (differentiated keratinocytes); (2) corneodesmosomes (proteinaceous rivets holding corneocytes together); and (3) lipids. The SC has a lamellar structure with well-structured lipid bilayers. The SC performs a barrier function thanks to the high content of proteins and lipids, i.e., ceramides (50%) containing phytosphingosine, fatty acids (10–20%, highly enriched in linoleic acid), and cholesterol (25%) [41].

NPs that do not have sufficient ability to penetrate the skin can be applied to the skin surface as photoprotective and antimicrobial agents (Ag-NPs, TiO_2_-NPs, ZnO-NPs, calcium carbonate CaCO_3_-NPs).

In recent years, more and more attention has been paid to the use of NPs as carriers in so-called transdermal drug delivery systems (TDDSs) [42]. The condition for the use of a TDDS is that the drug passes through the skin barrier, enters the bloodstream, and ensures therapeutic concentration. Carriers with NPs might not only improve the penetration of macromolecular compounds through the skin and improve their bioavailability, but also reduce their immunogenicity [43]. In addition to liposomes, solid lipid NPs (SLNs), polymer micelles, and inorganic NPs are used for this purpose, i.e., SiO_2_-NPs, Au-NPs, copper sulfide nanoparticles (CuS-NPs), and Fe_3_O_4_-NPs. Inorganic NPs have many advantages, such as good stability, the ability to modify the surface, and the ability to adjust size and shape, which determines their potential to penetrate the skin. Therefore, inorganic NPs are most often used to study the effect of the size of NPs on skin penetration. So far, TiO_2_-NPs and ZnO-NPs have been commonly used as reference NPs [44]. In another study, air pollutants such as soot and fine dust have been applied for this purpose [45]. The observed skin-aging effects were evidence of penetration of NPs through the skin.

As for the process of penetration, the skin is a porous barrier in which there are numerous semi-circular channels of a diameter between 0.4 and 36.0 nm. The primary barrier that NPs must overcome is the SC. There are three possible ways of NPs’ penetration through the skin: through the lipid matrix of the SC, through the pores of sweat glands (diameter: 60–80 µm), or through hair follicles, pilosebaceous pores (diameter: 10–70 µm), and sebaceous glands.

An interesting way to deliver the drug by the skin and transdermal route using NPs is through hair follicles, which are an important site of translocation and accumulation of NPs [46,47,48,49,50].

Topical drug delivery takes place across the SC via intercellular and intracellular routes or along the skin appendages via the transfollicular route (Figure 1). The SC is a lipidic acidic compartment, whereas the deeper layers form a more aqueous environment. Thus, small (<10 nm) lipophilic particles with a positive charge are able to passively penetrate through the epidermis into the deeper layers of the skin [51], whereas polar molecules diffuse into the deeper tissues. It can be assumed that the partition coefficients (log *P*) in the range from 1 to 3 provide effective skin absorption [52].

### 2.2. The Models of Skin Penetration

Skin penetration tests are necessary to confirm the performance of topical or transdermal products. For NPs to be therapeutically useful, they should penetrate the skin barrier, deliver their contents, and degrade without undesirable side effects. Penetration of NPs through a damaged skin barrier is indisputably easier. Penetration through a normal skin barrier is still being tested on various in vivo and ex vivo skin models. NPs, i.e., TiO_2,_ ZnO [53,54,55,56,57], quantum dots (QD) [51,58,59,60,61,62,63,64], and Au-NPs [57,58,59,60,62,64,65,66], were most often tested for penetration through the skin barrier. The FDA/EMA guidelines should be used in the design of the experiment; however, other methods are used in scientific experiments, which makes it impossible to compare the results.

Human skin is regarded as the so-called “gold standard” of membrane models in ex vivo skin penetration studies. Human skin is obtained from autopsy or plastic surgery resources. However, most studies on skin penetration by NPs are performed on animal skin (pig, mouse, rat, guinea pig, and rabbit), both in in vivo and in vitro conditions. It should be remembered that there are both structural and morphological interspecies differences in epidermal thickness and hair follicle density [67,68] that might affect the penetration of NPs. The results between the models of human skin and animal skin can differ because of the varied thickness of the SC, the number of hair follicles by area, the diameter of the follicles, and the amount, composition, and packing of the lipid matrix surrounding the corneocytes [69,70]. Caussin et al. [71] compared porcine and human SC lipids, which are arranged in a hexagonal lattice and a denser orthorhombic lattice, respectively. However, SC thickness appears to be comparable in the range of 21 to 26 μm, as well as hair follicle density, established as 20/cm^2^ for porcine ear skin and 14–32/cm^2^ for human forehead skin [72]. In turn, the dermis of a rat is thicker than that of humans. In addition, the skin of rats lacks subcutaneous fat and subcutaneous muscle, as well as the border between the papillary and reticular layers of the dermis.

Prior to the particle penetration test, the hairy skin is often subjected to depilation, cutting, or shaving, which may affect the barrier function of the skin. Senzui et al. [73] confirmed the effect of hair removal on obtained results by carrying out an experiment of the TiO_2_-NPs’ penetration through pig skin. He proved that hair removal enhances skin penetration by NPs, possibly through empty hair follicles. In in vitro skin penetration tests, the results are affected by the measurement method, i.e., flow cells through diffusion vs. static Franz diffusion cells [74], and the condition of the skin (damaged/undamaged) [75].

The ability to penetrate the skin also depends on the diffusion area and, more specifically, on the ratio of the volume of the preparation to the surface of the skin. An example is the comparison of the results presented in the works of Sonavane et al. [66] and Labouta et al. [76]. While Sonavane reported good penetration of citrate-stabilized Au-NPs into rat skin, Labout’s team presented no penetration by similarly sized NPs. The observed differences resulted from the almost 2-fold lower value of the ratio of the volume of the preparation to the surface of the skin (cm^2^) used by the second team.

Penetration testing can be performed using epidermal membranes, dermatomed skin, or full-thickness skin. There is a consensus that the use of the epidermis provides more reliable information and better simulates the in vivo situation. The presence of the dermis, when there is no continuous blood flow, creates an incorrect imitation of the skin barrier [77]. The use of the epidermis requires appropriate preparation, heating, or chemical treatment with enzymes, detergents, and salts, which unfortunately can cause structural damage and change the metabolic activity of the skin [78]. Particularly dangerous when separating the epidermis from the dermis is damage to the hair follicles. The lack of hair follicles affects the penetration of substances of hydrophilic nature and high molecular weight, which prefer the transappendageal route.

For the study of the role of the follicular route in the penetration of drugs through the skin barrier, it is recommended to use pig skin, due to its histological similarity and the number of hair follicles similar to the human model (20–30 hair follicles per cm^2^ of skin surface and hair density of 11–25 hairs/cm^2^, diameter 58–97 µm) [79]. Other models can be used for testing, such as EpidermFT™ (skin without hair follicles) or models of fibroblasts or keratinocytes with hair follicles [80].

The thickness of the full skin can be reduced by removing connective tissue and subcutaneous fat with a dermatome. The effect of membrane thickness on permeation rate has been studied many times [81,82,83]. In the example of testosterone, higher flow rates were confirmed for a thinner dermatomized skin thickness of 300–500 µm (2.82 to 5.39 µg cm^−2^ h^−1^) than for full-thickness skin of 700–900 µm (0.40 to 0 0.80 µg cm^−2^ h^−1^) [84]. This observation shows that for lipophilic molecules, the results obtained using epidermal membranes and even the dermatomed skin of full thickness may be overestimated due to the incomplete skin barrier devoid of part of the dermis, which actually hinders transport [85]. According to the OECD guidelines, skin penetration tests should be performed using dermatomed skin with a thickness between 200 and 400 μm. The use of cultured and reconstructed human skin models (e.g., constructed from keratinocytes) is not recommended, because they generate results incomparable with human skin [86]. Synthetic membranes are simple substitutes for human skin that provide better reproducibility of permeation results. For example, synthetic silicone membranes showed higher testosterone permeability (*K*_p_ = 176.4 × 10^−5^ and 777.6 × 10^−5^ cm h^−1^) compared to human skin (*K*_p_ = 7.6 × 10^−5^ cm h^−1^) and porcine (*K*_p_ = 31.7 × 10^−5^ cm h^−1^) [87]. Reconstructed skin is devoid of physiological receptors and constitutes a weaker barrier that is more permeable than human skin. Despite these imperfections, efforts are being made to create synthetic models. It is known that the properties of proteins, as well as the composition and proportions of fatty acids, especially supplementation with linoleic acid, are of great importance for ensuring the barrier function of the skin [88,89].

Another important aspect regarding in vitro research is tissue storage conditions. According to the guidelines proposed by the OECD, the EU Scientific Committee on Consumer Products, the US Environmental Protection Agency, and the International Program on Chemical Safety, the skin can be safely frozen without compromising its structural integrity at −20 °C. However, there is no agreement on the duration of storage. The International Program on Chemical Safety (IPCS) states that human skin can be stored for one year, whereas the US Environmental Protection Agency (EPA) allows storage for up to 3 months. There are large discrepancies in the published reports, e.g., Hewitt et al. [81] propose storage for 3 months, while Veryser et al. [82] proposed a maximum of 6 months. However, it should be emphasized that while frozen tissue can be used for permeation testing, it is not suitable for testing the metabolic activity of drugs. In addition, frozen skin after thawing should be adequately hydrated to be similar to fresh tissue [83].

### 2.3. Influence of Physical–Chemical Properties of NPs on Skin Penetration Efficiency

The way in which the physicochemical properties of NPs determine penetration, systemic translocation, and toxicity have all been studied in detail [16,52,90,91,92,93,94,95,96,97,98]. There is a consensus that the physicochemical properties of NPs, i.e., shape, chemical composition, stability, surface area, and charge, have a decisive influence on the interaction with the skin.

#### 2.3.1. Size Effect

The most important parameter determining the ability of NPs to penetrate the skin is the NPs diameter [99,100]. NPs smaller than 10 nm or smaller than 600 Da [101] have the ability to passively transfer through skin barriers and reach systemic circulation [102,103]. The outermost layer of the skin, i.e., the SC, is practically impermeable to larger particles [104]. Many studies describe a decrease in the permeability of NPs through the skin with the increase in their size [69]. The exception is TiO_2_-NPs, which cannot penetrate the skin layers, regardless of their size or shape [105]. The follicular route allows penetration into the hair follicles, which can become a reservoir of much larger NPs [99,106,107]. An experiment conducted on rats [107] showed that solid lipid NPs (SLNs) with different particle sizes have different abilities when penetrating different layers of the skin. When SLNs of approximately 100, 300, and 900 nm were exposed to rat skin, 300 nm SLNs were retained by the upper layers of the skin, while 100 nm NPs diffused through the hair follicles. The ability of larger NPs to reach the deeper layers of the skin via hair follicles has been confirmed by many researchers for both organic and inorganic NPs [108,109,110,111].

#### 2.3.2. Surface Charge Effect

Another important parameter, from the point of view of the permeability of NPs through the skin barrier, is the charge on the surface. In the absence of a surface charge, NPs can agglomerate due to the lack of electrostatic repulsion. The surface charge also determines the interaction of NPs with the skin. Negatively charged skin will repel negative NPs, causing them to aggregate on the surface. This reasoning is consistent with the work of Shanmugam et al. [112], who showed that cationic liposomes had a greater ability to penetrate the skin than anionic and neutral liposomes. Other studies reveal that Au-NPs functionalized with cell-penetrating peptides present a greater ability to penetrate the skin than negatively charged polyethylene-glycol (PEG)-functionalized NPs [113]. Current knowledge of Ag-NPs’ and Au-NPs’ cellular uptake was collected in a review by Talarska et al. [114]. Ryman-Rasmussen et al. [115] studied QDs of different sizes (4.6 nm, 12 nm core/shell diameter) and shapes (spherical, ellipsoid) and different surface coatings, i.e., neutral (PEG), anionic (carboxylic acids), or cationic (PEG-amine). The research was performed using the in vitro method using pig skin in flow diffusion cells. Once again, the results indicate that faster skin penetration for cationic and non-ionic QDs is related to the negative charge of the skin surface. Baspinar et al. [116] described that the penetration of a nanoemulsion containing prednicarbate with a positive charge owing to phytosphingosine was enhanced compared to the negatively charged formulation, thanks to myristic acid. Chitosan-coated NPs [117] showed a promising system for transdermal delivery of a lyophilic substance.

The aforementioned papers assumed that a positively charged topical formulation could lead to enhanced penetration due to an increased interaction with the SC, especially corneocyte components, which carry a negative charge [118]. Wu et al. [119] studied the disposition of charged NPs 100 nm in diameter (cationic amino-functionalized polystyrene, anionic carboxyl-functionalized polystyrene, and anionic poly-(L-lactide)) after their topical application on a porcine skin model. The fluorescent dye N-(2,6-diisopropylphenyl) perylene-3,4-dicarboximine as a model active compound was incorporated within each of NPs. The obtained results showed that the cationic NPs possessed the highest affinity for the negatively charged skin surface and delivered the greatest amount of the active agent into the SC. 

One should note the danger that positively charged NPs, due to easier penetration, may generate a greater toxic effect. Although most studies have confirmed easier penetration of positive and neutral NPs, there are also conflicting reports in this respect. Gillet et al. [120] studied the effect of the surface charge of liposomes on skin penetration, concluding that charged liposomes of phosphatidylcholine (150 nm) had better permeability of the model drugs betamethasone and betamethasone dipropionate through the skin compared to positively charged liposomes developed using stearylamine and neutral liposomes. Similar conclusions can be drawn based on the work of Lee et al. [121]. The authors studied the effect of the surface charges of gold nanorods (GNs) on skin penetration using a Franz-type diffusion cell (FDC), transmission electron microscopy (TEM), and inductively coupled plasma mass spectrometry (ICP-MS). The results showed increased permeability to the SC of the electron-dense dots of GNs compared to those with a positive charge (*p* < 0.01). Better diffusion coefficients of negative NPs crossing biological barriers are also confirmed by other studies [122,123].

#### 2.3.3. Hydrophobic/Hydrophilic Effect

Skin presents variable hydrophilicity and hydrophobicity in its different layers. Thus, effective penetration into the deeper layers of the skin provides the molecules with an amphiphilic character. There are several surface modification strategies for inorganic NPs, including physical adsorption, covalent bonding, layer-by-layer assembly, ligand exchange, and in situ polymerization [124]. Metallic NPs have usually already stabilized during the synthesis process. Stable dispersion in aqueous solutions is ensured, for example, by a hydrophilic citrate coating formed in the process of reducing the appropriate salt. In turn, coating the surface with oleic acid can make the surface hydrophobic. It is assumed that the hydrophobicity of the particle surface significantly improves skin penetration. This was demonstrated in the example of Au-NPs [76]. Au-NPs (15 nm) modified by cetrimide, creating a hydrophobic surface, had better penetration ability compared to citrate-stabilized NPs of the same size. Surface modification with lecithin, owing to the hydrophilic surface, reduced the penetration of even smaller Au-NPs (6 nm). It should be emphasized that hydrophobicity/hydrophilicity has a much smaller impact on skin penetration than differences in the size of NPs [125].

#### 2.3.4. Shape Effect

Another parameter that affects the crossing of the biological barrier by NPs is their shape, or rather distinct aspect ratios (length/width = ARs). The synthesis conditions determine the final product, which can have various shapes, from simple spherical and rod-shaped to rose-shaped NPs. Monteiro-Riviere et al. showed that spherical QD core shells had a greater ability to penetrate the skin than ellipsoidal QDs [115], while gold nanorods penetrated deeper than gold nanospheres [113].

Among the monodisperse mesoporous silica NPs (MSNs) which were sphere-shaped, short-rod-shaped, and long-rod-shaped, the latter were more easily internalized by cells compared with spherical NPs [126]. Xie et al. [127] observed that methyl-PEG-coated Au-NPs in the shape of stars, rods, and triangles differ in the efficiency of cellular uptake by RAW264.7 cells, which seems to be the worst for the star and the best for the triangles. An interesting observation was that, depending on the shape, NPs use different endocytosis pathways. Observations related to the influence of the shape of NPs on overcoming the biological barrier are described in the review by Wang et al. [128]. Fernandes et al. [113] studied the interactions between human and mouse skin and colloidal Au-NPs (15 nm) with different physicochemical properties. The penetration of Au-NPs through the skin was assessed using various techniques, i.e., ICP-OES, TEM, and a two-photon photoluminescence microscope (TPPL), which enabled the visualization of NPs’ migration within different skin substructures. These studies revealed that Au-NPs functionalized with TAT and R7 cell-penetrating peptides (CPP) accumulated in the skin in greater amounts than PEG-functionalized NPs and were able to penetrate deeply into the skin structure. The authors noticed that positively charged NPs penetrated the skin in larger numbers in comparison to their negatively charged counterparts. Furthermore, the rod-shaped NPs showed a higher accumulation in the skin compared to the spherical NPs. It turns out that the shape of NPs plays a greater role in permeation via the intercellular pathway, while permeation via the vesicular pathway is independent of shape [129].

#### 2.3.5. Chemical Composition Effect

NPs can be composed of various materials that interact with skin components and regulate the penetration of skin layers. There are lipid NPs, i.e., solid lipid NPs (SLNs), liposomes, nanostructured lipid carriers (NLCs), nanoemulsions, polymeric NPs, and inorganic NPs [130]. Thanks to low interfacial tension and good wetting properties, nanoemulsions ensure even deposition on the skin surface [27]. Lipid-based NPs form uniform layers on the SC and extend the residence time, thus promoting interaction with the skin layers and modulating its barrier properties. SLNs and NLCs combine the advantages of polymer particles, liposomes, and emulsions and can be used for dermal as well as transdermal drug delivery. SLNs contain lipid droplets that are fully crystallized. NLCs are modified SLNs in which the lipid phase contains both solid and liquid lipids. Both forms ensure high drug stability, controlled release profile, and high percentage of encapsulation.

*Solid Lipid Nanoparticles (SLNs)* (400–1000 nm) are based on solid lipids and emulsifiers. The physicochemical properties of SLNs depend on the lipid composition. Chantaburanan et al. [131] showed that the addition of secondary solid complex triglycerides to SLNs produced with cetyl palmitate resulted in higher ibuprofen encapsulation efficiency and improved sustained release capability. SLNs, as local drug delivery systems, are safe due to biocompatibility, biodegradability, and low toxicity. SLNs can be used as carriers for local delivery of high-molecular lipophilic molecules, ensuring their better efficiency in reaching the deeper layers of the skin. The adhesive properties provide even distribution of SLNs by creating a film on the surface of the skin. By interacting with the SC, they change its barrier properties and allow the drug to penetrate into the deeper layers of the skin [27]. Local drug delivery, thanks to SLNs, allows for increased drug deposition in areas such as hair follicles or sebaceous glands, which increases the therapeutic effect and prevents systemic negative effects [132]. This is confirmed by in vivo studies conducted on SLNs loaded with cyclosporin A and calcipotriol, which improved the treatment of lesions in psoriasis compared with free drugs [133]. Similar results were obtained with SLNs loaded with benzoyl peroxide (BPO) used in the topical treatment of acne [134]. The disadvantage of SLNs is the crystal structure of some solid lipids, high viscosity, and low physical stability, as well as the possibility of polymorphic transitions. Some limitations can be avoided by adding secondary solid complex triglycerides (Softisan 378; S378) to the matrix, e.g., cetyl palmitate [131]. The improvement of the penetration of SLNs through the skin is ensured by the addition of surfactants, which loosens the SC. SLNs are used in preparations for the treatment of dermatologic disorders, such as acne, psoriasis, androgenetic alopecia, hirsutism, ichthyosis, etc., as well as in anti-wrinkle cosmetic products. An example may be SLNs loaded with retinyl palmitate [135]. In the work of Chen et al. [136], SLNs were used as carriers to deliver powerful antioxidants to the skin, i.e., resveratrol, vitamin E, and epigallocatechin gallate (EGCG). In the work of Kelidari et al. [137], SLNs were used to deliver spironolactone (SP) to the skin. SP-loaded SLNs (SP-SLNs) were tested for drug release, skin penetration, and drug retention. SP-SLNs were characterized by a spherical shape with an average diameter, zeta potential, and trapping efficiency of 88.9 nm, −23.9 mV, and 59.86%, respectively. The analysis showed that the amount of SP penetrating the skin of the SP-SLN rat was almost double that of SP alone 24 h after administration.

*Nanostructured lipid carriers (NLCs)* are formed from a mixture of solid and liquid lipids such as oleic acid, triolein, copaiba oil, almond oil, etc. These systems are used to deliver dermal, transdermal, and vesicular drugs. The determining factor for the depth of delivery is the diameter of the NPs. Smaller particles reach the systemic circulation, while larger ones are located on the surface of the skin, and medium particles in the hair follicles [46]. A promising effect of using NLCs is the ability to improve skin hydration by creating a protective film in the SC and preventing water loss through the skin [138]. The use of NLCs for the purpose of vesicular drug delivery for the treatment of acne, hirsutism, and alopecia, i.e., the so-called androgenic skin diseases, seems to be promising as well. NLCs also act as nanocarriers for sunscreens and chemotherapeutics for wound healing, thanks to the possibility of extending the residence time of the preparation at the site of injury. It turns out that NLCs loaded with octyl methoxycinnamate (OMC) have better photoprotective properties than SLNs loaded with OMC [135]. NLCs prepared from lipids of natural origin are characterized by low toxicity. They can be used to encapsulate active pharmaceutical ingredients (APIs) and deliver them directionally through the skin. An example is thymol encapsulated in NLCs, with 107.7 (±3.8) nm composed of natural lipids [139]. The gel containing thymol NLCs exhibited anti-inflammatory and antipsoriatic activity on mouse models.

*Liposomes* are in the form of vesicles of an aqueous phase surrounded by one or more lipid bilayers, usually composed of phospholipids or cholesterol. Liposomes can be used for local administration of APIs, providing them with controlled release and retention in the skin, limiting systemic absorption. The above beneficial effects depend on the composition of the liposomes, particle size, lamellarity, fluidity, and occlusive properties [140]. Liposomes, especially transfersomes, are suitable for TDDSs [141], as they can even pass through the dermis [141]. Improving the elasticity of the lipid bilayer of liposomes, using surfactants or ethanol (ethosome), can increase penetration through the deeper layers of the skin. Most liposomes are applied together with surfactants to improve penetration [142]. Surfactants are mainly responsible for disorganizing the intercellular lipids of the skin, making it more permeable. In the case of ionic surfactants, such as sodium lauryl sulfate, this is done by interacting with keratin fibrils and non-ionic surfactants, such as Tween 80 and polysorbates, by dissolving lipids and interacting with keratin [143]. Ethosomes and transfersomes as ultradeformable vesicular carriers were used the percutaneous delivery of sulforaphane for the treatment of skin cancer diseases [144]. Sulforaphane is known to have antiproliferative effects against melanoma and other skin cancers. However, its poor permeability limits its clinical use. The applied ethosomes (<400 nm) consisting of 40% ethanol (*w*/*v*) and phospholipon 90G 2% (*w*/*v*) ensured an increase in the percutaneous penetration of sulforaphane and an improvement in the anticancer activity on SK-MEL 28 compared to the free drug [144]. In another study, elastic and ultradeformable liposomes (<300 nm) were used for delivering the anti-inflammatory and anticancer 3-(4′-geranyloxy-3′-methoxyphenyl)-2-trans-propenoic acid [145]. Nanotechnology has great potential in the treatment of actinic keratosis and malignant skin lesions, e.g., squamous cell carcinoma caused by UV radiation. Antiproliferative and antimitotic drugs are used to treat these diseases. Paclitaxel-loaded ethosomes [146], in an in vitro study, improve paclitaxel penetration and increase antiproliferative activity compared to the free drug. The vesicular colloidal carriers, ethhosomes and transfersomes loaded with linoleic acid, were applied in the therapeutic treatment of hyperpigmentation disorders [147]. The liposomes were prepared using the lecithin component and ethanol and sodium cholate. Experimental findings showed that both carriers were accumulated in the skin membrane model and can be applied for topical delivery of linoleic acid. As with other nanostructures, smaller liposomes, e.g., small uni-lamellar vesicles (SUV) of around 70 nm, can penetrate deeper compared to multi-lamellar vesicles (MLV) with a diameter of around 300 nm. Liposomes > 600 nm remain on the surface of the SC. There are several mechanisms for the penetration of liposomes into the lipid lamellae of the SC and through the epidermis [27].

*Niosomes* are vesicular systems with a single- or multi-layer spheroidal structure formed from the connection of amphiphilic molecules, i.e., Spans and Tweens. As non-ionic surfactants, they are non-toxic and biocompatible to enhance the penetration of local drug delivery systems. Tweens, however, prove to be more effective in the TDDS as compared to Spans. A similar effect is obtained by reducing the amount of cholesterol in niosomes [148]. Penetration of the skin by niosomes is strongly related to the type of surfactants, mainly their hydrophilic–lipophilic balance (HLB) number. Lipophilic surfactants (HLB = 9–10) cause greater drug retention in the skin. Cationic niosomes, as a rule, enhance penetration into the skin, which is negatively charged due to surface lipids, in contrast to anionic or neutral niosomes [149]. The mechanism of skin penetration by niosomes includes reduction in transepidermal water loss (TEWL), and the second is fusion or adsorption of vesicular drug delivery systems to the surface of the skin. Niosomes can loosen the SC, thanks to the presence of terpenes, which makes it more permeable and bioavailable for encapsulated drugs to pass through the skin [150]. The addition of ethanol or essential oils improves the elasticity of the bubbles and causes fluidization of the lipids of the SC, which improves the solubility of drugs in the stratum corneum. Ethanol allows a reduction in the size of niosomes by modifying the surface charge and steric stabilization. The essential oils in the follicular membrane also improve elasticity and improve drug delivery across the skin. The addition of clove, eucalyptus or lemon essential oils to niosomes has been studied for transdermal delivery of felodipine [151]. Spherical niosomes (279–345 nm) (Span 60 and cholesterol) were used. Thanks to the presence of essential oils, the fluidization effect of the membrane and the improvement of drug release were obtained. One of the niosomal formulations is a preparation containing 5-aminolevulinic acid (ALA) for photodynamic therapy (PDT) of skin cancers [152]. Niosomes were prepared with a Span 60–cholesterol mixture, ethanol and various boundary activators, diethyl phosphate (DCP), and sodium cholate (SC). Niosomes were more effective in ex vivo permeation and penetration of ALA through human skin than the drug solution. Other niosomal preparations were obtained from a mixture of Span 60 and Tween 60 for encapsulation of the plant antioxidant ellagic acid (EA), which in the water form is poorly soluble and poorly permeable to the skin. Nanosomes were prepared with Span 60 and Tween 60 and various solubilizers, i.e., PEG 400, propylene glycol (PG), and methanol (MeOH) [153]. The obtained niosomes had the shape of spherical multilayer vesicles of 124–752 nm. The in vitro study showed that the penetration of EA depended on, among others things, the type of added solubilizers. In order to improve the penetration of proteins and peptides into niosomes, additives such as poly-oxyethylene ether and diacyl glycerides are added [154].

*Nanocrystals* are systems with a size of 1 to 1000 nm formed from solid particles of drugs [155]. In order to avoid aggregation, the particles are suspended in ionic (sodium lauryl sulfate), non-ionic (Poloxamers, Tweens), or polymer stabilizers (hydroxyl propyl methylcellulose, polyvinyl alcohol, polyvinyl povidone, hydroxyl propyl cellulose). The advantage of these carriers is a very high drug-loading capacity, thanks to which it is possible to obtain a high therapeutic concentration at the target site [156]. In turn, nanonization allows, by reducing the size of particles, to increase the surface area of nanocrystals, which is important in the case of penetration of poorly hydrated skin. Nanocrystals improve the delivery of poorly water-soluble drugs through the skin [156]. The literature describes many examples of improved skin penetration by drugs applied in the form of nanocrystals compared to conventional forms, e.g., nanosuspension of lutein [157] and dexamethasone-nanocrystal-loaded ethyl cellulose nanocarriers [156]. The increase in the penetration of nanocrystals is the result of an increase in the concentration gradient, which enhances the passive diffusion of APIs through the skin layers [155,158].

*Polymer NPs* are colloidal nanocarriers with dimensions below 1000 nm. Polymer NPs are characterized by very good adhesion, which ensures a long residence time on the skin, which is why they have gained recognition for topical application [159]. There are basically two types of nanopolymers, i.e., nanocapsules and nanospheres. Nanocapsules have a bubble structure with an oil component, and nanospheres have no oil in their composition. Natural and synthetic polymers have rather low toxicity and good biocompatibility; however, to improve the APIs’ permeability and extend the contact time, interpenetrating polymeric networks (IPNs) such as hydrogels have been introduced. Hydrogels are nanocarriers suitable for transferring hydrophilic molecules such as peptides, proteins, and oligonucleotides. The advantage of hydrogels is primarily better stability compared to nanoemulsions and suspensions. It should be noted that synthetic polymers are superior to natural polymers in terms of purity, i.e., uniformity of composition. Polymer NPs can be made of poly-ε-caprolactone, chitosan, poly (lactide-co-glycolide), poly-(ε-caprolactone)-block-poly (ethylene glycol), poly (butyl cyanoacrylate), poly (ethyl cyanoacrylate), ethylcellulose, cellulose acetate phthalate, and a fatty-acid-conjugated poly (vinyl alcohol) [160]. The chitosan nanogel (370.4 ± 4.78 nm) formulation containing anti-diabetic drugs glibenclamide and quercetin was prepared by ionic gelation [161]. The percentage of cumulative drug release through skin showed favorable results. Other types are dendritic polymers (dendrimers), i.e., poly (amidoamine) and poly (propylene imine) dendrimers with a core–shell structure, which enhance the penetration of APIs through the skin by interacting with skin lipids and denaturing keratin proteins. As in the case of other nanoparticles, the permeation efficiency is determined by the size, charge, and functionalities on dendrimers [162]. Polymer NPs can be used as carriers of lyophilic APIs and UV-protective ingredients in topical applications due to the possibility of retention on the skin surface [159]. This was confirmed in the study of passive penetration of human skin by fluorescent dyes 5-dodecanoylaminofluorescein and Nile Red, as model lipophilic compounds, enclosed in tyrosine-derived nanospheres [163]. Nanospheres (50 nm) were formed from a copolymer of PEG, oligomers of suberic acid, and desaminotyrosyl tyrosine alkyl esters. In turn, the penetration of deeper layers of the skin is ensured by hydrophilic nanogels [164].

*Inorganic NPs* are very stable with wide-range functionality. Skin penetration in this case is size-dependent, similarly to other types of NPs [66]. Many authors indicate the possibility of aggregation of inorganic NPs that are applied on the surface of the skin. This phenomenon is highly unfavorable in terms of penetration [69]. The surface of inorganic NPs can be modified, which significantly changes the effectiveness of skin penetration. An example may be Au-NPs with a surface modified with a hydrophobic coating, thanks to which they can penetrate deeper [74,165]. Silica NPs can be used as carriers for transdermal drug delivery. They have many advantages, i.e., adjustable pore size and size and ease of functionalization via reactive silanol groups on the surface [166]. Human skin blocks the penetration of silica NPs larger than 75 nm [167]. NPs with a hydrophobic surface penetrate into the deeper layers of the skin [125]. Negatively charged and smaller silica NPs (20 nm) are more toxic [168]. It turns out that amorphous silica NPs can penetrate the skin barrier and induce an immunomodulatory effect [169]. A relationship between the size of the NPs (300–1000 nm) and adjuvant activity was proven using an atopic dermatitis model induced by intradermally injected *Dermatophagoides pteronyssinus* (*Dp*) mite antigen in NC/Nga mice. Reducing the silica particle size increased interleukin IL-18 and the production of thymic stromal lymphopoietin (TSLP), leading to systemic inflammatory T helper (Th) 2 responses and exacerbation of allergic skin lesions. Another example of carriers used for transdermal drug delivery are Au-NPs. Numerous studies confirm that the size of Au-NPs and their shape have a large impact on the rate of skin penetration [66,170]. A solid-in-oil dispersion of Au nanorods can be used to enhance the transdermal delivery of protein, as well as skin vaccination [171].

### 2.4. Methods to Improve the Penetration of NPs through the Skin

For an effective therapeutic effect, NPs must pass through the SC barrier [172]. Transdermal delivery of particularly hydrophilic molecules is hindered by the lipid layer of the epidermis. To generate a perforation in the SC, various technologies are used, such as microneedles, cavitation ultrasound, microdermabrasion, electroporation, and thermal ablation.

#### 2.4.1. Exposure to UV Radiation

Exposure to UV radiation is used to improve skin penetration by inorganic NPs. Studies have shown that UVB not only increases the penetration of NPs through the skin, but also changes the biology of skin cells. For example, the uptake of QDs in keratinocytes, primary melanocytes, and related cell lines is increased after initial exposure to UVB. Increased penetration of QDs as model NPs is the result of skin damage on the surface of the SC around the hair follicles [173].

#### 2.4.2. Local Hyperthermia

Mild local hyperthermia can be used for the percutaneous application of vaccines. Mild local hyperthermia (42 °C) enhances transdermal (HET) immunization. This is considered a novel strategy employing the application of antigens along intact skin, resulting in detectable antigen-specific immunoglobulins (Igs) in serum [174]. For example, mice transdermally immunized with diphtheria toxoid generated an antibody response. Therefore, it can be assumed that local hyperthermia increases the transport of high-molecular-weight NPs and antigen-labeled NPs [175].

#### 2.4.3. Iontophoresis 

Iontophoresis, thanks to the use of direct current, improves the transdermal penetration of drugs through the skin [176]. The combination of nano-drugs and iontophoresis was first described in 1996. Electromigration and electro-osmosis appearing due to the application of a low-level electric current (≤0.5 mA/cm^2^) are responsible for enhancing the passage of drugs throughout the skin. It should be emphasized that positively charged NPs are preferable for iontophoresis. 

The deposition of triclosan by triclosan-loaded cationic nanospheres (261.0 ± 15.1 nm) combined with iontophoresis was 3.1-fold greater in comparison to the application of the triclosan solution by passive diffusion [177]. Ethylcellulose/Eudragit^®^ RS NPs loaded with dexamethasone (105 nm) and positively charged (+37 mV) have shown the potential to control the release and penetration of corticosteroids into the skin, thereby reducing the side effects of corticosteroids [178]. Results show that electric fields of different intensities (3.2~9.8 V/cm) can improve the permeability of a transdermal chip system for the delivery of Au-NPs [179]. Penetration of Au-NPs into the human SC via the intercellular route was proven.

#### 2.4.4. Dermaportation and Sonophoresis

Penetration of NPs through the skin is improved by the use of pulsed electromagnetic fields (PEMF) [180] and ultrasound at low frequency (20–100 kHz) [181], known as dermaportation and sonophoresis, respectively. The penetration enhancement mechanism is related in these cases to the formation of transient pores in the SC. 

#### 2.4.5. Mechanical Permeation Enhancement

In the in vivo study conducted by Gulson et al. [182], humans were exposed to sunscreens containing 19 nm and >0 nm ZnO-NPs, with the aim to determine if zinc (Zn) can be absorbed through the undamaged skin. Stable isotope (68) Zn tracing allowed dermally absorbed Zn to be distinguished from Zn already present in the blood compartment. It appeared that the majority of applied (68) Zn was not absorbed, and the amount of tracer detected in blood after the 5-day application period was ~1/1000 that of total Zn in the blood. So far, mechanical treatments, i.e., skin flexion and massage, have not been reported to affect the penetration of NPs into the skin. However, regarding the research of Lademann et al. [183], it appears that hair movement simulated by massaging the skin may facilitate the penetration of medium-sized NPs (~400–700 nm) into the hair follicles. Additionally, a study by Gratieri et al. [184] performed using the gold standard of human skin showed that QDs passed into the deeper skin layers (DSLs) after massaging (5–10 min) of tape-striped skin. Thus, the use of NPs as drug delivery vehicles seems to be most effective for partially damaged skin.

TDD is made possible by the use of microneedles. Microneedles (MNs) are micron-sized needle protrusions 10–2000 μm high and 10–50 μm wide that painlessly penetrate the skin. However, frequent use of MNs may lead to skin damage and even inflammation.

Mechanical methods of enhancing penetration include tape-stripping and dermabrasion. These methods involve complete or partial removal of the SC [185,186,187,188,189]. Dermimage is used in cosmetic procedures to increase skin permeability to hydrophilic preparations. Few studies describe the penetration of NPs through dermabraded skin [58,59,60]. Tapeless skin has an altered permeability to NPs [51,58,59,60,61,64,190]. The 2009 Gopee study showed that nail-shaped QDs coated with neutral-charge PEG (CdSe/CdS core/shell, 37 nm) and negatively charged QDs coated with spherical dihydrolipic acid (CdSe/ZnS core/shell, 15 nm) accumulate in the mouse liver after topical application in dermabraded but not intact SKH-1 hairless mouse skin [60]. Neutral QDs showed greater accumulation compared to negatively charged ones, further indicating the influence of surface charge.

#### 2.4.6. Chemical Permeation Enhancement

In order to improve the efficiency of NPs’ penetration through the skin, chemical permeation enhancers can also be used. The results showed that oleic acid (OA), ethanol (EtOH), and oleic acid-ethanol (OA-EtOH) were all capable of enhancing the transdermal delivery of ZnO-NPs by increasing the intercellular lipid fluidity or extracting lipids from the SC [191]. Among the tested chemical enhancers, dimethyl sulfoxide (DMSO) could induce the penetration of hydrophilic (citrate-stabilized) gold colloid [192].

#### 2.4.7. Thermal Ablation

Thermal ablation is used to generate perforations in the SC by microheaters, radiofrequency, or laser. Photothermal NPs, e.g., Au-NPs, absorb near-infrared (NIR) light (650–900 nm), causing resonance and the transfer of thermal energy to the surrounding tissue. The photothermal effect can be used, for example, in the ablation of tumor cells [193] or to increase drug permeability [194]. Another class of photothermal NPs is the semiconductor copper monosulfide NPs (CuS-NPs). They have an advantage over Au-NPs because, in addition to lower cost, the absorption wavelength of CuS-NPs is independent of the dielectric constant of the surrounding medium [195].

## 3. Methods Applied to Investigate the Skin Penetration by NPs

### 3.1. Visualization

Detection and quantification of NPs is a great challenge for analytical chemistry. The techniques applied to monitor skin penetration by NPs should offer low detection limits due to the trace concentration of NPs penetrating the skin barrier. Most studies were conducted using inorganic NPs. Scanning electron microscopy (SEM), TEM, fluorescence microscopy, and confocal and multiphoton microscopy are useful for the qualitative visualization of these NPs. The penetration of NPs as a carrier of bioactive compounds into the skin can be carried out using confocal laser scanning microscopy (CLSM). The confocal and multiphoton laser scanning microscopy enables the achievement of three-dimensional visualization of NPs’ distribution in skin layers via optical sectioning [69]. These techniques offer high image resolution as well as magnification of objects up to two million times. Determining the presence or absence as well as concentration of NPs in biological tissue has been enabled by multiphoton microscopy (MPM). MPM, in combination with fluorescence lifetime imaging microscopy (FLIM), can identify fluorophores with overlapping spectral properties. The use of MPM and FLIM to visualize the disposition of NPs and QDs in the skin was described in a review article [196]. For the understanding of nanomedicines’ in vivo behavior, imaging techniques such as nuclear imaging by positron emission tomography (PET) and single-photon emission computed tomography (SPECT) are highly attractive because of high sensitivity and non-invasive quantification. The nuclear imaging techniques can provide information about pharmacokinetic parameters, biodistribution profiles, or target site accumulation of nanocarriers [197]. Freeze-fracture electron microscopy (FFEM) can visualize the effects of interactions occurring between NPs and the skin structures. Samples should be appropriately treated by freezing and subsequently fracturing under a high vacuum [198].

### 3.2. Quantification of NPs and Structural Changes in Skin

Quantification of elemental composition requires other analytical methods, such as inductively coupled plasma optical emission spectroscopy (ICP-OES), ICP-MS, and atomic absorption spectroscopy (AAS). The disadvantages of these techniques are spectral interferences from trace elements and the possibility of analyzing atoms or ions originating from biological materials not just from the NPs themselves. In turn, high-pressure liquid chromatography (HPLC) with diode array DAD or mass spectrometry (MS) detection provides for the quantitative analysis of medicinal products.

Fourier Transform infrared (FTIR) spectroscopy can be used not only to evaluate drug stability, but also the organization of the SC, especially the lateral lipid organization of the intercellular lipid matrix. FTIR modification, namely attenuated total reflectance FTIR (ATR-FTIR), enables the measurement of the SC in vivo to generate a penetration profile of APIs as well as to evaluate their effects on lipid organization after topical application. Differential scanning calorimetry (DSC) is used to answer the question of how drug penetration through the skin influences the lipid bilayers. Another method used to study the effects of NPs on the lamellar organization of lipids in the intercellular matrix of the SC is small-angle X-ray diffraction (SAXD) [199].

### 3.3. NPs Physicochemical Characteristics

The NPs physicochemical characteristics cover particle size (nm), shape, and zeta potential (mV) evaluation. Light scattering can be useful for assessing these properties. The particle size, the zeta potential, and size distribution can be measured by wet laser diffraction sizing known as dynamic light scattering (DLS) using a Zetasizer. The specific surface area is measured by Brunauer, Emmett, and Teller’s method, known as the BET, through the evaluation of gas adsorption, typically krypton or nitrogen, by a monolayer.

## 4. Nano-Drug Therapy

Nanomedicine uses various types of engineered nanoparticles for dermal and transdermal drug delivery. In addition, there are so-called nanoparticle carriers, i.e., colloidal systems with a diameter < 500 nm, such as nanoemulsions, nanostructured lipid carriers (NLC), liposomes, niosomes, and others used as the TDDS.

The advantage of the TDDS is precise drug deposition, increased drug stability, and its controlled release [200]. Owing to topical application, the TDDS ensures avoidance of hepatic first-pass metabolism as well as the gastrointestinal tract. Furthermore, the amount of drug administered can be lower, permitting the use of a relatively potent drug without the risk of system toxicity. Nanoparticle therapies include (i) skin cancer imaging and therapeutic targeting, (ii) immunomodulation and vaccine delivery [201], and (iii) antimicrobial agents and wound healing (Figure 2).

### 4.1. The Transdermal Drug Delivery

#### 4.1.1. Skin Cancer Imaging and Targeted Therapy

The most dangerous skin cancer is metastatic melanoma [202]. Nanotechnology has been used both in its diagnosis (magnetic NPs, QDs, Au-NPs) and therapy [203,204,205,206,207,208,209].

Tumor-selective diagnostic probes must meet safety criteria, i.e., low nanotoxicity in vivo, no particle residues in the reticuloendothelial system, favorable distribution kinetics, efficient renal clearance, prolonged circulation time, and adequate tumor penetration. In molecular imaging of skin cancers, Au-NPs are used due to the possibility of coupling with many detection methods based on optical absorption, electrical conductivity fluorescence, atomic and magnetic force, and Raman scattering [210]. QDs offer other possibilities. They are usually covered with an anionic oligomeric phosphine envelope. After absorbing the radiation, they emit constant and stable fluorescent radiation with a wavelength depending on the size of the particles [211]. Cornell dots (C dots) received the first recommendation approved by the FDA. Core–shell silica NPs containing Cy5 dye (>650 nm) coated with methoxy-terminated PEG chains (PEG ~0.5 kDa) have also been described as potentially useful for selective tumor targeting in animal models of melanoma [209]. The neutral PEG coating prevented uptake by other cells. The use of bifunctional PEGs enabled the attachment of cRGDY peptide ligands targeting the ανβ3 integrin, additionally labeled with 124I positron-emitting radionuclide, with the aim of 3D visualization with PET. Despite rapid clearance, silica NPs had the advantage of being non-toxic and biodegradable [212,213]. Currently, the product is cleared for clinical trials for tumor targeting and lymph node mapping.

Nanomaterials are also being tested for the treatment of melanoma. Lipophilic and polymeric NPs are commonly used to deliver substances into the skin [214,215,216], mainly because of their easy degradation compared to insoluble NPs that accumulate in the skin. Compared to standard chemotherapeutic agents, which are cytotoxic to healthy cells, nanodrugs enable the selective delivery of higher drug doses to cancer cells [58,59,217,218,219,220,221,222,223,224,225,226]. An example is Au nanospheres modified with antibodies, which are used in phototherapy for selective photothermolysis of the tumor [227]. 

Other NPs are also used to attach homing ligands. Au nanocages [228], Au nanospheres [229], QDs [230,231], and polymeric liposomes [220,232] have been described in melanoma metastasis studies. The ligands associated with NPs target receptors that are overexpressed on melanoma cells. One such receptor is the melanocortin 1 receptor [228,229,233,234,235]. The mechanism of functioning of these receptors is mediated by the G protein and consists in signal transduction by increasing the concentration of cAMP in the neuroplasma and mobilizing intracellular calcium reserves [236]. MC1R is a classical receptor for α-MSH. It is expressed in melanocytes of the skin, but also in keratinocytes, fibroblasts, endothelial cells, and antigen-presenting cells (APCs). It is a key protein involved in melanogenesis. MC1R, after binding to α-MSH, initiates a complex signaling cascade that leads to the production of a black-brown pigment—Eumelanin. The attachment of peptide agonists or antagonists to NPs targeting the melanocortin 1 receptor does not present sufficient cellular specificity. In addition to melanocytes and melanoma cells, other cells also express the melanocortin 1 receptor [237,238,239,240]. Another receptor target is the sigma 1 receptor, delivering c-Myc small interfering RNAs to B16F10 melanoma tumors using a mouse model [220].

#### 4.1.2. Immunomodulation and Topical Vaccine Delivery

The skin contains a network of skin APCs, such as Langerhans cells (LCs), epidermis and dermal dendritic cells (DCs), dermal macrophages in the dermis, and other infiltrating cells, such as neutrophils, monocytes, inflammatory DCs (CD206+), and plasmocytoid DCs. The immune response is generated mainly by DCs and LCs, which are key regulators in the immune system [241,242,243]. For example, the skin is known to be responsible for sensitization to allergens [244,245,246]. Typical LCs are present in the suprabasal epidermis and rectal skin, foreskin, and body mucosa [247]. LCs are mainly recognized by the expression of the human leukocyte antigen (HLA)-DR and CD1a. Currently, the marker for LC identification is langerin [248]. CD1a+ cells are known to concentrate in the funnel epithelium of hair follicles [249]. NPs can easily accumulate in hair follicles, especially after mechanical stimulation [249,250]. The amount and depth to which the NPs penetrate along the alveolar duct depends on their size. However, it should be borne in mind that cleaning the hair follicle openings and reducing the barrier function in healthy skin has the potential to trigger an inflammatory response [251]. From the hair follicles, NPs diffuse into the perifollicular tissues and are taken up by LCs (CD207ţ) and dendritic cells (CD205ţ), which migrate to the lymph nodes [252,253]. 

Using an in vivo mouse model, it was shown that NPs, e.g., carbon nanotubes, have an immunostimulating effect. They are responsible for inducing macrophage activation, antigen-specific and non-specific T cell proliferation, cytokine production, and induction of antibody responses to ovalbumin [254,255]. TiO_2_-NPs injected subcutaneously in NC/Nga mice, after simultaneous exposure to mite allergen, intensify the development of skin lesions similar to atopic dermatitis (AD) [256]. In turn, combined exposure of the skin to TiO_2_-NPs and UVR exacerbated the symptoms of atopic-like dermatitis in DS-Nh mice [257]. In this case, UVR induces a defect in the skin barrier [258] and an increase in the penetration of NPs through the SC. 

NPs are capable of carrying an antigen [259], acting as an adjuvant [260]. Thus, nanotechnology has the potential to modulate the immune system [261,262,263,264,265] and deliver vaccines through the skin [266,267]. The nanocarrier-based model vaccines appeared to be effective in animals, e.g., the transcutaneous application of a model HIV-p24 particle-based vaccine, which triggered serum and mucosal antibodies as well as cellular immune responses [268]. The most likely route of penetration for topical vaccination is through the hair follicles due to the abundance of immune-competent cells around the root sheath and the sebaceous gland. Jung et al. [269] reported that other routes of penetration, such as interalveolar penetration, play a minor role. The first study presenting the potential of non-invasive, transfollicular vaccination using NPs without compromising the SC barrier was published by Mittal et al. [201]. NPs were prepared from polymers poly(lactide-co-glycolide) (PLGA) or chitosan-coated PLGA (Chit-PLGA) with polyvinyl alcohol as a stabilizer. Ovalbumin (OVA) was used as a model antigen. The mean size of OVA-loaded NPs was ca. 170–180 nm, and a negative surface charge of −24.8 ± 0.89 mV was measured for OVA-loaded PLGA or a positive surface charge of 20.2 ± 1.05 mV for OVA-loaded Chit-PLGA. Blank PLGA and Chit-PLGA NPs without incorporated antigen had slightly smaller particle sizes and elevated the overall surface charges compared to the loaded NPs. Malleable NPs composed of organic substances such as lipids, proteins, and polymers can be the matrices for antigens used in vaccines. For instance, ISCOMs composed of phospholipids, cholesterol, saponifiers, antigens [270], or virosomes, viral hybrid liposomes, are used in HBV and HPV vaccines [271,272]. Vogt et al. [249] proved that a maximum of 40 nm nanoparticles may be efficiently used to transcutaneously deliver vaccine compounds via the hair follicle into cutaneous APCs. Skin penetration routes of NP-based vaccines are presented in Figure 3. Excellent reviews concerning nanoparticulate carriers used as vaccine adjuvant delivery systems have been recently published [60,61,273,274,275].

#### 4.1.3. Gene Therapy

The morphogenesis of the epidermis involves, among others things, miR-203, which supports the transition from proliferation to differentiation during epidermal stratification, miR-205, which maintains the proliferative capacity of basal cells in the nascent epidermis, and miR-214, which is involved in the embryonic development of hair follicles. In turn, miR-146a and miR-21 are relevant in the context of psoriasis. Several review articles have been written on the role of miRNAs in skin development [276,277]. 

The highly proliferative stem cells, located in the bulge region of the hair follicle, are capable of differentiating into other cell types like keratinocytes and melanocytes. Therefore, these cells are attractive targets for gene therapy of hair and skin genetic disorders. Gene-suppressing oligonucleotides, such as antisense DNA oligonucleotides and RNA interference with small interfering RNAs (siRNA) and microRNAs (miRNA), are designed to be complementary to the genetic targets. 

Recently, RNA interference was used for the treatment of pachyonychia congenita (PC). This genetic skin disease is characterized by mutations in one of four keratin genes (KRT6a, KRT6b, KRT16, or KRT17) [278]. The methods of the transdermal delivery of oligonucleotides that mimic or inhibit miRNA function taking part in the mechanism of skin diseases are constantly being developed. 

NP carriers are able to deliver DNA and si-RNA molecules to the stem cells as well as protect them from degradation. It was described that nanoparticle carriers are able to deliver the lacZ reporter gene, as well as DNA and oligonucleotides to hair follicle progenitor cells [279] in animal models, and the gene encoding β-galactosidase to follicular stem cells at wound borders. The carriers are usually equipped with penetrating peptides or viral vectors, such as HIV-based vectors [280], or a retrovirus, which was used to deliver the tyrosinase-gene pLme/SN in the therapy of albinism and hair growth disturbances [281]. 

Au-NPs provide an attractive and applicable scaffold for the delivery of nucleic acids. Ding et al. [282] prepared the review focusing on the use of covalent and noncovalent Au-NP conjugates for applications in gene delivery and RNA interference technologies. Transdermal siRNA delivery via liposomes, transfersomes, etosomes, transetosomes, and SECosomes has been reported [277,283].

### 4.2. NPs for Topical Application

In case of poor ability to penetrate the skin, NPs are used for topical application. Left on the surface of the skin, they are used to heal wounds, treat skin inflammation, or prevent damage caused by UV light. Calcium carbonate (CaCO_3_) and calcium phosphate Ca_3_(PO_4_)_2_-NPs have the potential to prevent nickel (Ni) allergy due to their ability to trap metal ions in the cation exchange process [284]. Several valuable reviews of NPs in dermatology have been published in recent years [285,286].

#### 4.2.1. NPs as UV Protestant (against Photoaging and Photocarcinogenesis)

The ultraviolet radiation (UVR) spectrum we are exposed to on earth includes UVA (320–400 nm), UVB (290–320 nm), and UVC (100–290 nm). UVA reaches the earth through the ozone layer, UVB is partially blocked, and UVC is completely filtered out by the ozone layer. UVB radiation is responsible for sunburn and direct DNA damage (formation of thymidine dimers). UVA penetrates into the DLS and generates ROS that damage proteins, lipids, and nucleic acids (guanine). ROS are responsible for the destruction of collagen in the skin and the reduction in its synthesis [287]. UV radiation is therefore dangerous for the skin, as it causes not only the desired tanning, but is responsible for sunburn and subsequent consequences in the form of photoaging [288,289]. Photoaging of the skin causes dryness, wrinkles, seborrheic keratosis, as well as loss of elasticity and slower wound healing. Unfortunately, regular UV exposure and repeated sunburn cause hyperalgesia, which is associated with skin cancer. 

The Himba society inhabiting the northern part of the Kunene region in Namibia uses natural red Namibian ocher in the form of a preparation called Otjize as a skin and hair beautifying cream. Red ocher has been found to be a solar heat IR reflector and an effective UVA- and UVB-blocking agent. It should be noted that the Himba community has an extremely low rate of skin cancer, although due to weather conditions they are exposed to the standard 5% UV (300–400 nm) all year round. Morphological and crystallographic studies of the red ocher pigment have confirmed the presence polymorphic forms of iron (III) oxide and iron (III) oxy-hydroxide, namely α-Fe_2_O_3_ and lepidocrocite (γ-FeOOH) nanocrystals [290]. Ocher Otjize additionally showed antimicrobial efficacy against *Escherichia coli* (*E. coli*) and *Staphylococcus aureus* (*S-aurus*). The antibacterial activity of the Himba Otjize’s red ocher relies on three potential mechanisms: a photocatalytic mechanism, an ROS mechanism, and a bacterial cell surface interaction with iron (III) oxide nanocrystals. Besides the effective UV filtration, Otjize minimizes the skin overheating by reflecting back at least half of the solar heat in the IR region. 

The American Academy of Dermatology and other medical scientific groups, such as the American Cancer Society, recommend topical UV protection in the form of sunscreen. UV-absorbing agents should accumulate in the upper skin layers to form a dense light-absorbing layer and additionally guarantee water resistance. The development of nanotechnology has allowed the replacement of physical and chemical (octyl methoxycinnamate, oxybenzone, octocrylene, and luteolinone) sunscreen agents without the risk of side effects. TiO_2_-NPs and ZnO-NPs are among the strong means of physical protection against UV radiation. Mixtures of TiO_2_-NPs and ZnO-NPs are even more effective, protecting against both UVA and UVB [291]. 

It is known that under the influence of UV radiation, free radicals are formed, which cause cellular damage, e.g., collagen responsible for skin aging. Topical in situ skin delivery of antioxidants can be used to protect the skin from free radicals’ attack. Plant extracts, rich in phenolic compounds, can effectively protect the skin against photoaging. Bucci et al. [292] developed nanoberries, ultradeformable liposomes made from soybean phosphatidylcholine and sodium cholate carrying blueberry (*Vaccinium myrtillus*) extract. Nanoberries of about 100 nm appeared to be nontoxic and effective for protecting the skin from UV. Shetty et al. [293] developed PLGA polymeric NPs (diameter 90.6 nm, zeta potential of −31 mV encapsulated with flavonoid-morin possessing antioxidant and UVR protection properties). 

For protection against UV radiation, NPs are used: (i) polymer, (ii) metallic, and (iii) fullerenes.

*Polymer NPs:* Benzophenone-3 (BZ-3) is a molecular filter that protects the cosmetic against deterioration of its quality under the influence of light and protects the skin against the harmful effects of solar radiation. The ingredient is included in the list of radioprotective substances and, in principle, does not pose a health risk, apart from its contact allergic and photoallergic potential. It turns out that the nanoencapsulation of such traditional UV filters in organic nanocarriers, such as biocompatible and biodegradable polymers (poly(lactic acid) (PLA), poly(glycolic acid) (PGA), their copolymer PLGA, poly(caprolactone) (PCL), N-(2-hydroxypropyl)-methacrylate copolymers, and poly (amino acids)), improves their retention in the skin, stability, and radiation-blocking effect [294,295]. Nanocarriers particularly preferred are PCLs, as they provide reduced penetration through the skin and thus are able to maintain the preparation on the skin surface for a long time without penetrating into the deeper layers of the epidermis and dermis [296]. Hyperbranched polyglycerol (PLA-HPG) NPs are an example of a nanocarrier with strong adhesive properties. In animal studies, padimate-*O* (PO) encapsulated in PLA-based nanocarriers was tested [297]. Such a formulation had more than 20 times the ability to protect against UV radiation than the UV filter alone. It is a natural polymer that has the ability to absorb UV radiation like lignin. Lee et al. [298] developed a skin protection product containing light-colored lignin (CEL) NPs from rice husk and organic UV filter BIOTHERM Lait Solaire Hydratant (SPF 15), achieving a synergistic protective effect against exposure to UVA radiation.

Metallic NPs TiO_2_ and ZnO have been known for decades as effective UV filters. Their use, however, has been limited, as they also reflect light in the visible spectrum, creating the undesirable effect of a white grainy coating on the surface of the skin. NPs of these metal oxides, TiO_2_-NPs and ZnO-NPs, with dimensions of 40–60 nm, thanks to the absorption, reflection, and scattering of UV light, are more effective and transparent, thanks to which they are accepted by consumers and can be used in sunscreen products. These NPs have been approved by the FDA as filters to protect the skin from photoaging [299]. The photoprotective effect is the result of physical phenomena such as absorption, scattering, and reflection of UV radiation [294]. ZnO-NPs protect against UVA and UVB radiation, while TiO_2_-NPs only protect against UVB. Metallic MPs are not biodegradable and remain on the surface of the skin for a long time, ensuring its protection [300]. Studies show that these NPs cannot penetrate into the DLS [105,191]. Increased penetration may occur in the presence of chemical enhancers that have increased the intercellular lipid fluidity of the epidermis, i.e., ethanol, oleic acid, ethanol, and oleic mixtures [191]. Photoinduced disaggregation may have a similar effect on percutaneous penetration [301]. It turns out that after a few minutes of lighting, the hydrodynamic diameter of TiO_2_ aggregates decreases from ~280 nm to ~230 nm, which causes an almost three-fold increase in penetration. An interesting finding was the increase in the photostability of ketoprofen (KP) under the influence of TiO_2_-NPs [301]. ZnO-NPs can, under certain conditions, release Zn^2+^ ions, which, as an essential chemical element, have many beneficial biological properties. Particularly beneficial, from the point of view of skin protection, is the participation of Zn in enzymatic processes. Zn, by inhibiting nicotinamide adenine dinucleotide phosphate oxidase (NADPH), protects the skin against ROS [302]. Somewhat worrying are the data collected in 2006–2010 by James et al., which showed that a small amount of Zn from NPs can enter the circulation and trigger an immune response [303]. Topically applied ZnO-NPs suppress allergen-induced skin inflammation but induce vigorous IgE production in the atopic dermatitis mouse model [304]. On the other hand, TiO_2_-NPs are insoluble, and in the form of agglomerates, they cannot penetrate. It should be emphasized that Ti has a low toxicity and has been used in dentistry for years to produce orthopedic implants.

In order to increase biocompatibility and improve photoprotection, various modifications of metallic surfaces with organic molecules with defined activity are used. The product of such modifications is nanocomposites. In a study by Jo et al. [305], ZnO-NPs (70 nm) were coated with chitosan or niacinamide, which are known as cosmetic skin-lightening agents. The ability of surface-coated ZnO-NPs to protect against UV radiation was expressed as a sun protection factor (SPF). It turned out that ZnO-NPs coated with niacinamide showed an improvement in SPF compared to ZnO-NPs and ZnO-NPs coated with chitosan. The combination of NPs and antioxidants enhances free radical scavenging activity. In the work of Shetty et al. [293], a plant flavonoid morin with antioxidant and UV-protective properties was used for this purpose. The polymeric NP loaded with morin in the presence of ZnO-NPs and TiO_2_-NPs ensured the delivery of morin to the epidermis, preventing it from penetrating into the systemic circulation. The combination of NPs used showed antioxidant activity and protection against UV radiation.

In 2019, the first report on the photoprotective properties of peptide-modified NPs was published [306]. ZnO-NPs (60 nm) were modified with a secondary amphipathic peptide M9 (CRRLRHLRHHYRRRWHRFRC). Modification of the ZnO-NPs’ surface was aimed at obtaining dispersion stability and protection against aggregation. Aggregation of NPs is an unfavorable phenomenon, because the sunscreen should be evenly distributed over the surface of the skin. The modification of the NPs with the M9 peptide imparted a positive charge on the surface of the ZnO-NPs, due to the sequence of arginines and histidines, which prevented aggregation and ensured skin compatibility. Other benefits of the modification turned out to be the lack of toxicity, limited percutaneous penetration, and extended residence time in the stratum corneum. Improved retention of modified NPs in the upper epidermal layer resulted from the interaction of the peptide with skin lipids and corneal-desmosome proteins. The peptide coating protected the NPs from releasing the free Zn^2+^ ions. The photoprotective properties of the modified NPs were tested on an animal model. In an in vivo study, SKH-1 mice irradiated with UVB showed lower genotoxicity, lower levels of oxidative stress induction, less response to DNA damage, and lower immunogenic potential for peptide-modified ZnO-NPs compared to unmodified NPs.

UV-photoprotective properties are also presented by NPs obtained as a result of the so-called ‘green synthesis’. An example may be a study from 2022, confirming the effectiveness of a natural water extract of *Hoodia gordonii* as a means for the synthesis of single-phase CeO_2_ nanocrystals with an average diameter of ~5–26 nm with 4+ electronic valence [307]. The average value of the reflectivity varies from 45 to 63%, depending on the annealing temperature. The authors declare that UV selectivity is the result of dominant absorption in the UV spectral region. An NPs-CeO2 was characterized by photostability coupled with low formation of harmful reactive oxygen species (ROS), like singlet oxygen and hydroxyl radicals, that damage skin cells.

*Fullerenes:* Fullerenes (C60) are known as free radical scavengers [308], as they have many conjugated double bonds in the LUMO (lowest unoccupied molecular) orbital, enabling them to accept an electron. One C60 molecule neutralizes as many as 34 methyl radicals [309]. Fullerenes encapsulated in polyvinylpyrrolidone (PVP) with the so-called “radical sponge” have been studied as a protective component of human skin keratinocytes against UVA radiation [310]. In general, fullerenes dispersed in water are safe in mammals. It is worth noting that their water-soluble derivatives have greater ROS-scavenging activity and better membrane-tropic functions through a greater affinity for nucleic acids and proteins [311].

Carbon fullerene and its polyhydroxylated derivatives as free radical scavengers contribute to the improvement of the skin’s antioxidant capacity [308]. In a study from 2010 [312], the effectiveness of new fullerenols (C(60)(OH)(6–12):LH-F, C(60)(OH)(32–34) 7H(2)O:HH-F, and C(60)(OH)(44)8H(2)O:SHH-F) for UVA or UVB damage to human skin was demonstrated. Irradiation of HaCaT cells (UVB) causes many harmful effects i.e., increase in oxidative stress, formation of cyclobutanepyrimidine dimers, and chromatin condensation, all of which were inhibited by SHH-F. The authors reported suppression of these damages by one of the new derivatives, namely SHH-F.

Fullerenes have the ability to protect against photodamage to the skin barrier. Murakami et al. [313] describe the photoprotective effect of UVB inclusion complexes of fullerenes with polyvinylpyrrolidone (PVP/fullerenes). The inhibition of keratinocyte proliferation after UVB irradiation of human skin was restored in the presence of PVP/fullerenes. In addition, the use of PVP/fullerenes protected against the consequences of UVB, i.e., reduced expression of transglutaminase-1 and transepidermal water loss.

#### 4.2.2. Antimicrobials and Wound Healing

Wound healing is not just a local problem. It is known that some chronic diseases, such as obesity, diabetes, and atopic dermatitis, may impede this process [314]. Most hard-to-heal wounds are infected by strains of bacteria that are resistant to antibiotics, such as methicillin-resistant SA (MRSA) [315]. Silver compounds, i.e., metallic silver, silver nitrate, and silver sulfadiazine, have been used for a long time to treat burns, wounds and skin infections of bacteria, viruses, and fungi, which have an effect on both Gram-positive and negative bacteria [316]. However, silver ions in high micromolar doses (1 to 10 µM) are toxic to mammals [317]. 

As it turns out, silver in the form of NPs exhibits better antimicrobial activity compared to silver compounds [318]. Antibacterial and antifungal effects have been confirmed [78,319], along with poor skin penetration by Ag-NPs. This may be due to the large surface area, which is able to provide better contact with microorganisms. 

Ag-NPs have been found to be useful in the production of clothing [320,321], food storage containers [322], washing machines [124], soap, and surgical masks [323]. Wound dressings containing a nanocrystalline silver coating are available on the market, e.g., Acticoat (SMITH & NEPHEW), Silverlon (KIK), and Actisorb Silver 220^®^ wound dressings. Products containing nano Ag are highly effective for wound healing (nano Ag) [324,325]. 

Ag-NPs and Ag^+^ present different action mechanisms. While Ag^+^ ions binding to DNA cause inhibition of bacterial enzymes [326], Ag-NPs cause damage to the bacterial wall and cytoplasmic membrane of bacteria [327]. The mechanism of action of Ag-NPs on bacteria is not fully understood. Proteomic studies show that exposure of bacteria to Ag-NPs causes an increase in the level of three precursors of outer membrane proteins. Ag-NPs have also been reported to affect the *cis–trans* isomerization of membrane unsaturated fatty acids. All this indicates that the action of Ag-NPs is primarily related to the destruction of the structure, and thus the permeability of bacterial membranes [328] (Figure 4).

Chronic exposure to Ag-NPs can lead to silver accumulation, which is manifested by a bluish graying of the skin (Argyria-6.4 g of colloidal silver per a year) [329]. Exposure to AgNPs via inhalation was linked to numerous neurodegenerative disorders [330]. Ag-NPs, thanks to their antimicrobial properties, are found in a wide range of products that come into contact with the skin (clothing, creams). However, the safety profile for Ag-NPs is still under investigation. The exposure of damaged skin to Ag-NPs is of great concern [331,332]. The toxicity of Ag-NPs has been confirmed in in vivo and in vitro studies in pigs. Toxic effects in the form of epidermal edema and dermatitis and epidermal hyperplasia have been observed [333]. In vitro studies have shown cytotoxicity and increased production of cytokines in keratinocytes on skin devoid of a protective barrier, e.g., on open wounds. Korani et al. [334] reported the dermal toxicity of Ag-NPs in an animal model of the male guinea pig. Dermal toxicity was shown to be dose-dependent. For animals treated in the low doses of 100 µg/mL, a reduction in the thickness of the epidermis and dermis was observed, with a slight increase in inflammation of Langerhans cells. In animals that were exposed to 10,000 µg/mL Ag-NPs, collagen fibers ruptured. Studies performed using a pig skin model [333] showed that after topical application of Ag-NPs (20, 50, and 80 nm), areas of focal inflammation were observed. Ag-NPs were detected on the surface and in the upper stratum corneum, as well as in cytoplasmic vacuoles of human epidermal keratinocytes (HEKs). 

Currently, more and more attention is paid to Ag-NPs modified or functionalized through covalent and non-covalent bonds to ligands and thiol groups, which have even stronger antibacterial properties. The modified surface of Ag-NPs by antibiotics is able to refine the effectiveness of their antimicrobial activities. The conjugation of Ag-NPs with medicaments can be directly carried out by ionic/covalent bonding or physical absorption. For example, Morales-Avila et al. [335] proved that the functionalization of Ag-NPs with the cationic peptide ubiquicidin 29–41 (UBI) causes a significant increase in antibacterial activity against *E. coli* and *P. aeruginosa.*

TiO_2_-NPs also exhibit antibacterial properties. After irradiation (UVR), NPs act as a photocatalyst in the peroxidation reaction of a polyunsaturated phospholipid in the lipid membrane of bacteria [336]. On the other hand, chlorhexidine-loaded nanocapsule-based gel (Nanochlorex) provides prolonged antibacterial activity, the effectiveness of which is comparable to 60% 2-propanol [337]. Another example of products that support wound healing is nitric-oxide (NO)-releasing nanoparticles. A NO-releasing platform can be prepared using silane-hydrogel-based nanotechnology [338]. NO-releasing NPs (10 nm) are made of components such as tetramethylorthosilicate, polyethylene glycol, and chitosan. 

Recently, the photocatalytic and antibacterial properties have been described for a new orchestration of X-CuTiAP (X-ethylenediamine (en), triethylenetetramine (trien), ethanolamine (ETA) and dimethylamine (DMA)) nanospheres, and Au-NPs synthesized using Bauhinia tomentosa Linn extract by Gnanamoorthy et al. [339,340]. The authors studied the activity of the obtained NPs against *Staphylococcus Aureus* (Gram-positive), *Pseudomonas aeruginosa* (Gram-negative) pathogens, and *Candida albicans* (fungal) pathogens compared with Gram-negative *Escherichia coli* (*E. coli*) and Staphylococcus aureus, respectively.

Borate Bioactive Glasses (BBGs) are interesting biodegradable, bioactive materials for wound-healing applications. Currently, BBGs are exploited for not only soft tissue (wound healing, nerve, and muscle regeneration), but also hard tissue (bone regeneration, drug delivery for osteomyelitis, or osteonecrosis treatment) engineering applications [341]. In the case of BBGs, the wound-healing process improves due to the ionic dissolution products of BBGs [342], especially boron, which is important in different stages of wound healing. It stimulates angiogenesis [343], takes part in the synthesis of the extracellular matrix, and stimulates the secretion of collagen and proteins; it stimulates HUVEC proliferation and migration associated with the MAPK signal pathway [344], promotes keratinocyte migration [345], and upregulates the vascular endothelial growth factor (VEGF) [346]. Moreover, boron is also an antiseptic agent that aids the wound-healing process [347]. Mirragen microfibers made of boron 13–93B3 glass are an example of a commercial product approved by the U.S. Food and Drug Administration (FDA) in 2016. 

BBGs can be also incorporated in polymeric scaffolds usually enriched with different dopants (copper, zinc, gallium, and silver), enhancing antimicrobial properties [343,348]. However, it should be remembered that this action is dose-dependent, and too low a level cannot inhibit bacterial growth [349], whereas too high a dose could harm fibroblast and keratinocyte cells or reduce angiogenesis [350].

#### 4.2.3. Treatment of Psoriasis

Psoriasis is an inflammatory skin disease that attacks the immune system. The causes of skin changes include abnormal interactions of keratinocytes with T lymphocytes [351]. Psoriasis therapy includes local, systemic (methotrexate, cyclosporine) and phototherapy (narrowband UVB (NB-UVB), excimer laser/lamp (targeted phototherapy), and psoralen plus UVA (PUVA). One of the first topical treatments for psoriasis was coal tar, which is now banned in Canada and the European Union. UV light has the ability to delay the growth of skin cells and remove psoriatic plaques. The recommended PUVA therapy for clinical treatment uses this effect, which is a combination of the light-sensitive natural compound psoralen plus UVA (UVA) radiation.

The main drugs for topical treatment of psoriasis are corticosteroids, which inhibit the release of phospholipase A2 and act directly on deoxyribonucleic acid (DNA) and inflammatory cytokines [352]. Corticosteroids act synergistically with retinoids, salicylic acid, and vitamin D analogs. Retinoids (tazarotene) act on retinoic acid receptors and retinoid-X-receptors, initiating gene expression’s modification of inflammatory cytokines, and thus inhibiting keratinocyte proliferation. Psoriasis therapy also includes immunosuppressants such as calcineurin inhibitors (tacrolimus, pimecrolimus), suppressing the production of the inflammation’s potent cytokine mediators (IL-2, IL-6, and IFN-γ) and stimulating suppressor T cells [353]. Dithranol is also used topically to reduce epidermal hyperproliferation and inflammation [354].

Unfortunately, topical delivery of traditional drugs is not effective due to poor skin penetration and possible skin irritation caused by high doses of drugs. A breakthrough in the treatment of psoriasis was the discovery of IL-17 and IL-23 pathways involved in the pathogenesis of psoriasis and the introduction in 2004 of biological drugs containing recombinant proteins (monoclonal antibodies and fusion proteins), which inhibit the activity of inflammatory cytokines [355]. An example is Secukinumab-blocking interleukin-17 (IL-17) cytokine related to the T-helper-17 (TH-17) pathway. Currently, there are many biologics for the treatment of psoriasis such as etanercept, guselkumab, infliximab, ustekinumab, etc.

To overcome the SC, various types of NPs have been developed, mainly lipid-based, as drug delivery systems. Thanks to the amphiphilic properties, the lipid envelopes reduce the toxicity and increase the bioavailability of APIs of a lipophilic and hydrophilic nature.

#### 4.2.4. Treatment of Vitiligo

Vitiligo is an acquired idiopathic skin disease that is characterized by depigmentation mainly in exposed areas of the body. The cause of the appearance of white spots on the skin is apoptosis or damage to melanocytes. The etiology of vitiligo is not fully understood, but autoinflammation and oxidative stress are considered to be the most important factors. Endogenous killer and inflammatory dendritic cells are known to be hyperactive in patients with vitiligo. As a result of the autoimmune response, autoantibodies and cytokines are secreted, i.e., INF-γ,14–16 CXCL10,14,17,18 TNF-α, IL-6, and IL-17 [356], which contribute to the destruction of melanocytes. The first-line drugs in the treatment of vitiligo are anti-inflammatory and immunosuppressive corticosteroids (betamethasone dipropionate, clobetasol dipropionate, and mometasone furoate) and calcineurin inhibitors (tacrolimus, pimecrolimus). In order to remove excess ROS and hydrogen peroxide from the epidermis, antioxidants are also used, i.e., polypodium leucotomos, vitamin E, vitamin C, and minocycline. Phototherapy, including narrow-band ultraviolet (NB-UVA) and psoralen ultraviolet A (PUVA), is also effective in inducing repigmentation [357]. Currently, special systems have been developed for the treatment of vitiligo, i.e., liposomes, polymer nanoparticles, microspheres, solid lipid nanoparticles and nanofibrous structures, and more advanced liposomes such as invasomes, transferosomes, ethhosomes, and niosomes. Transfersomes get their excellent elasticity due to surfactants such as sodium cholate, spans, and tweens. Invasomes, on the other hand, gain transdermal permeability due to the presence of soy phosphatidylcholine, ethanol, and terpenes.

#### 4.2.5. Baldness Treatment

NPs have been shown to increase drug penetration into the hair follicles by acting as a deposit for sustained drug release; therefore, they are effective in the treatment of hair growth disorders such as androgenetic alopecia and alopecia areata. Minoxidil, a pyridine derivative, is mainly used for topical treatment of androgenetic alopecia. The drug works by widening the blood vessels around the hair follicles. The problem of the treatment is irritation and contact dermatitis. The elimination of side effects is ensured by the use of nano-systems that allow the drug to be delivered to the hair follicles.

Examples of drug-loaded NPs for topical applications in the treatment of different skin diseases are summarized in Table 1.

## 5. Nano-Dermocosmetics

Reviews of the latest advancements of nanotechnology in cosmetics and cosmeceuticals have been recently written [35,407,408,409,410,411]. 

### 5.1. Nanoparticles as Anti-Aging Agents

NPs in anti-aging products are used as a therapeutic agent to slow down aging and as a means of protecting the skin from external stresses such as radiation and pollution [130]. Both biodegradable and non-biodegradable NPs have been studied for anti-aging applications. Biodegradable NPs are useful for encapsulating active substances, which enables sustained release and thus an extended therapeutic effect. In turn, non-degradable NPs, i.e., TiO_2_ and ZnO, act as protection against skin photoaging.

The inorganic NPs in these cosmetic products act primarily as effective sunscreens. For their synthesis, simple techniques can be used that allow obtaining NPs of very small sizes. It is also possible to modify the surface of the NPs, which extends the range of their applications. However, long-term use may raise concerns about toxicity [412,413]. Currently, the cosmetics industry uses less toxic biocompatible NPs [414,415]. Commercially available are Ag-NPs [416], platinum–palladium (Pt-Pd) [417], and Au-NPs [418], which have anti-wrinkle, skin-whitening, or antioxidant properties owing to strong reducing agents such as rutin and *Panax ginseng* (*P. ginseng*), used for the surface modification. TiO_2_-NPs and ZnO-NPs are used mainly as UV filters [419]. CeO_2_-NPs, in addition, exhibit antioxidant and antiapoptotic properties [419]. In the case of ZnO-NPs coated by chitosan, besides UV protection, a skin-whitening effect was observed by Schneider and Lim [299]. In turn, Aditya et al. [306] reported anti-inflammatory activity of peptide-coated ZnO-NPs.

Organic NPs as nanocarriers not only increase the stability of the supplied antioxidants, vitamins, or peptides, but also ensure better penetration into the skin. No objections have been raised regarding their long-term use, thanks to their proven biocompatibility and biodegradation, which does not cause immunological reactions [203,264].

#### 5.1.1. Lipid NPs

The composition of lipid NPs usually includes such lipids as phosphatidylcholine, cholesterol, and lecithin, found in skin tissues, which ensures excellent biocompatibility. The leading systems for the skin delivery of anti-aging active substances are various types of nanostructured lecithin gels [420], from classic liposomal hydrogel to modified forms (transferosomal, ethosomal, pro-liposomal, phytosomal), and vesicular phospholipid gel (VPG).

Lipid NPs can be in the form of micelles, SLNs, nanostructured lipid carriers (NSCs), and nanovesicles such as liposomes, niososomes, etasomes, transfersomes, and cubosomes. Lipid NPs possess the ability to carry both hydrophilic and hydrophobic bioactive molecules, providing them with high drug loading, stability, and excellent permeation through the skin layers. Several formulations have been described that are used in anti-aging products. Bi et al. [421] described liposomes (93 nm) that have been used to deliver vitamin D3. Liposomes enhanced the therapeutic effect of vitamin D3 and ensured the stability and protection of the skin against photoaging, increasing the production of new collagen fibers. The results indicate that liposome retention in the skin was 1.65 times greater compared to the vitamin D3 solution.

Coenzyme Q10 (CoQ10) is a powerful antioxidant used, for example, to protect against aging. Its lipophilicity and high molecular weight make it difficult to deliver it by topical application. The development of a liposomal (<200 nm) formulation of soy phosphatidylcholine (SPC) and alpha-tocopherol (vit. E) improved the local bioavailability of CoQ10 (*p* < 0.05) and doubled its accumulation in the skin [422]. Another proniosomal (PN) gel formulation of CoQ10 was prepared on the basis of soy lecithin and cholesterol [423]. In this case, the spherical vesicles formed from the hydration of proniosomal gel exceeded 1 µm. Despite this, CoQ10 PN showed better skin permeation, almost two-fold higher compared to conventional gel. The effectiveness of skin photoaging treatment was confirmed by measuring the level of antioxidant enzymes, i.e., superoxide dismutase (SOD), catalase (CAT), and glutathione (GSH).

Both liposomes (LPs) and ethhosomes (ETHs) improve penetration of drug molecules through the skin. In Yücel’s study [424], ETHs were found to be more effective than LPs in the transdermal delivery of rosmarinic acid. The study was confirmed by measuring the antioxidant activity and the inhibitory effect of the preparations on collagenase and elastase enzymes. The measured size range of the etosomal formulation was 138 ± 1.11 nm.

In turn, dispersions of alpha-lipoic acid (ALA) cubosomes, obtained using poloxamer gel (P407) as a carrier, have shown efficacy in the treatment of aging skin [425]. The product was tested on volunteers, resulting in a reduction in facial wrinkles in the area of the eye socket and upper lip, as well as an overall improvement in skin texture and color.

Cosmetic preparations containing extracts of rice (*Oryza sativa* L.) bran trapped in niosomes by supercritical carbon dioxide proved to be very effective. The use of preparations by volunteers in a monthly treatment improved the state of hydration, brightening, thickness, and elasticity of the skin [426]. It should be noted that rice bran extracts are a rich source of antioxidant compounds, including ferulic acid, γ-oryzanol, and phytic acid.

Thanks to the nanosize, better bioavailability of the active substance is obtained, and thus the effectiveness of the anti-aging effect. For the transdermal delivery of antioxidant enzymes, i.e., Cu,Zn-SOD, and CAT, carriers composed of various mixtures of soybean phosphatidylcholine (SPC/NaChol), mainly in the form of lipid bilayers, have been developed [427].

#### 5.1.2. Nanoemulsions

Nanoemulsions are kinetically stable colloidal systems. In the case of nanoemulsions, droplet sizes range from 20 to 500 nm. The encapsulation of nanoemulsions of bioactive compounds ensures ease of application and increases their solubility, controlled release, and penetration through the skin. Topical formulations are usually O/W emulsions prepared using emulsion inversion point or through high-pressure homogenization [428]. These types of oil-based nanoemulsions are used in anti-aging products. Oil phases are usually of natural origin, such as sunflower oil, tea tree oil, soya lecithin [429,430], olive oil, or cosmetic oils such as Eutanol G. According to Gupta [428] the amount of API in the oil phase is 80–100 mg/g, while the oil phase is typically about 15–20 wt. of the entire formulation. Nanoemulsions also include surfactants, usually non-ionic surfactants such as Tween 20, Tween 80, polyvinyl alcohol, or natural products like sucrose esters and cyclodextrins [431]. A common addition to facilitate application to the skin is the addition of a cross-linking agent to convert the formulation into a gel, such as carbopol 940 [432], glycerol, or PEG.

An anti-wrinkle nanoemulsion containing the hydrophilic molecule acetyl hexapeptide-8 (AH-8) was developed by Hoppel et al. [433]. Another example of a tea-tree-oil-based nanoemulsion is a preparation for transdermal delivery of fish protein hydrolysates (FPH) [434].

Nanoemulsions based on Compritol ATO containing an additional two-component mixture of surfactants were used as carriers for applying resveratrol to the skin [435]. A high drug load was achieved, which ensured the effectiveness of (i) antioxidant activity confirmed by the study of the activity of antioxidant enzymes (CAT, GSH, SOD), (ii) anti-inflammatory activity, confirmed by the study of anti-inflammatory markers interleukin 6 (IL-6), interleukin 8 (IL-8), and rat nuclear factor kappa B (NF-κB), and (iii) the anti-wrinkle test (matrix metalloproteinase (MMP-1) and granulocyte macrophage colony stimulating factor (GM-CSF)) after UVB irradiation.

#### 5.1.3. Nanoparticles of Precious Metals, i.e., Pd, Pt, and Au

Noble metal nanoparticles are characterized by strong catalytic activity in many chemical reactions, such as hydrogenation, hydration, and oxidation. This property results from the large surface area and high proportion of atoms on the surface of NPs [436]. In addition, noble metal NPs are believed to be powerful antioxidants [437,438]. Pd is known to prevent the oxidative degradation of Pt. Already in 1915, Hideyo Noguchi and Saburo developed a solution of Pd and Pt NPs used as a medicine against many chronic skin diseases, i.e., burns, frostbite, and urticaria, as well as other diseases such as pneumonia, acute gastritis, chronic gastritis, and rheumatoid arthritis [439]. Many years later, the therapy was recreated by Dr. Ishizuka, who developed PAPLAL, a mixture of Pd and Pt NPs [440]. PAPLAL was patented as an antioxidant against superoxide anions and hydroxyl radicals [441] in the Japanese Patent Office (Patent No. 3411195, 2003). Recently, Elhusseiny and Hassan confirmed the anticancer and antimicrobial activity of the complex of Pd-NPs and Pt-NPs [439]. The anti-aging effect of the PAPLAL complex (Pd:Pt; 2.7:1) applied transdermally in a mouse model was demonstrated by Shibuyai et al. [417]. Au-NPs, unlike Au in the bulk state, can absorb light and convert it to heat, acting as miniature thermal scalpels to remove, for example, cancer cells [69].

In the Cao et al [442] study, percutaneous permeation of Au from Au nanosheets as well as a cream containing Au nanosheets was systematically investigated using guinea pigs. Au from both preparations was demonstrated to be able to permeate into the skin in a time-dependent manner, but could not enter the systemic circulation. The main permeation route of Au nanosheets was through hair follicles. It was revealed by synchrotron radiation X-ray fluorescence (SRXRF) imaging. Unfortunately, both the extracted Au nanosheets as well as the Au nanosheets embedded in cosmetic creams inhibited the growth of hair. It has been observed that the expressions of hair growth marker proteins (CD34, ALP, and KRT19) were downregulated after exposure to the cosmetics containing Au nanosheets. The extracted Au nanosheets, in contrast to the cosmetic cream, were nontoxic to keratinocytes and skin fibroblasts.

Unfortunately, metallic NPs synthesized by chemical methods require the use of toxic-reducing and stabilizing substances (hydrazine hydrate, sodium borohydride, DMF, and ethylene glycol), which are adsorbed on the NPs. The above phenomena reduce the biocompatibility of nanomaterials and limit their use in medicine and cosmetology [443]. Therefore, natural methods of synthesis are gaining more and more popularity. This is the so-called green synthesis or biogenic or phytochemical synthesis using extracts from plants, yeasts, fungi, and bacteria. NPs obtained in this way are stable and less toxic compared to chemical synthesis products. Importantly, the bioactive components of the reducing extract, e.g., vitamins, alkaloids, carotenoids, polyphenols, fats, carbohydrates, and proteins, are adsorbed as stabilizing factors in the formation of NPs [444]. Ag-NPs (468.7 nm) loaded with phytochemicals have so far been used in anti-aging applications. Radwan et al. [416] showed that Ag-NPs stabilized with ethanolic *Eucalyptus camaldulensis* bark extract, the main component of which is rutin, reduced cell senescence and apoptosis in a human melanocyte cell line (HFB-4). The authors also confirmed a significant decrease in the activity of elastase, collagenase, and tyrosinase enzymes. Another example is the synthesis of Ag-NPs (87.46 nm) using *Symphytum ofcinale* leaf extract [11]. The anti-aging effect of S-AgNPs was studied using HaCaT keratinocyte cells treated by Ag-NPs after UVB irradiation. The authors emphasized photoprotective properties of Ag-NPs as indicated by the inhibition of matrix-degrading enzymes metalloproteinase-1 and pro-inflammatory cytokines IL-6 and increasing the expression of procollagen type 1 in keratinocytes.

Jimenez et al. [415] obtained Au-NPs in a green synthesis process using *P. ginseng* berry extract. Non-toxic to human skin fibroblasts, Au-NPs have a high potential for cosmetic applications, thanks to the ability to retain moisture and mitigate damage caused by oxidative stress. In addition, Au-NPs significantly reduced melanin content and suppressed tyrosinase activity in α-MSH-stimulated B16BL6 cells. 

#### 5.1.4. Cerium Oxide Nanoparticles (CeO_2_-NPs)

UVA radiation is particularly dangerous for the photoaging of human skin. The reason for this phenomenon is the formation of ROS in the epidermis and dermis. CeO_2_-NPs have been shown to have a protective effect against skin photoaging due to their ability to scavenge free radicals [445]. The antioxidant activity of CeO_2_-NPs is similar to that of the antioxidant enzymes SOD and CAT. Li et al. [419] studied the effect of CeO_2_-NPs on human skin fibroblasts (HSF) irradiated with UVA. The authors confirmed that CeO_2_-NPs may reduce the production of pro-inflammatory cytokines, intracellular ROS, β-galactosidase activity, and phosphorylation of c-Jun N-terminal kinases (JNKs) after exposure to UVA radiation.

#### 5.1.5. Anti-Aging Polymeric Nanoparticles

Many polymer compounds are used for the production of NPs, i.e., PLA, PGA, PLGA, polyvinyl alcohol (PVA), and PCL. Polymer NPs serve as drug delivery vehicles and are rarely used in anti-aging cosmetic products [446]. In cosmetic products, polymeric NPs play a photoprotective role. Among the few examples that report the use of polymeric NPs in anti-aging products is curcumin encapsulated in silk NPs (700 nm) (silk/curNPs). Curcumin is a natural antioxidant isolated from turmeric (*Curcuma longa*), thus it is functional for anti-aging. Yang et al. [447] showed that synthesized silk/cur NPs had an effect on markers of aging (P53, P16, HSP70 gene expression, and β-Galactosidase activity). In the study, the percentage of ß-galactosidase in rat bone marrow mesenchymal stem cells (rBMSCs) decreased from 36% to 25.7% after treatment with silk/cur NPs.

Nanocarriers present in cosmetic products along with active substances are listed in Table 2, which was developed on the basis of review articles [286,448,449,450].

## 6. NPs Protection against Pollution

Air pollution, in addition to the obvious threat to human health, affects the condition of the skin. It is known that these pollutants generate a state of oxidative stress by reducing antioxidant enzymes in the skin epidermis and the content of antioxidants, such as ascorbic acid, tocopherol, or glutathione, and increasing the secretion of pro-inflammatory cytokines. Exposure to PM therefore results in skin diseases such as acne, atopic dermatitis, psoriasis, and cancer that develop over time [451,452]. The resulting cellular damage to the skin is strongly associated with skin aging [453,454]. Rembies et al. [455] collected information on the effect of pollutants on the skin in his review article. The authors took into account particulate matter (PM) with sizes of about 2.5 and 10 μm and polycyclic aromatic hydrocarbons (PAHs), volatile organic compounds (VOCs), nitrogen and sulfur oxides, carbon monoxide, ozone, and heavy metals. The described skin changes associated with exposure to air pollution include many changes, including changes in lipid composition, collagen, elastin, melanin, lipids, proteins, pH value, and others. [455]. Changes in the composition affect the rate of secretion of the serum and hydration.

D-biotin, or vitamin H or B7, known as coenzyme R or Biopeiderm, is used in cosmetics to moisturize and smooth the skin. D-Biotin also supports the formation of collagen and elastin, which is why it is sometimes used in the treatment of skin diseases, e.g., eczema and acne [456]. Regeneration of skin affected by air pollution (reduction in erythema, improvement of elasticity and hydration) was achieved using biotin encapsulated in a water-in-oil-in-water W/O/W multiple emulsion system [429].

Eupafolin (6-methoxy 5,7,3′,4′-tetrahydroxyflavone) is another natural ingredient isolated from *Phyla nodiflora*, belonging to the family *Verbenaceae*. The alcohol extract of this plant has anti-inflammatory properties and is used in traditional Chinese medicine. In a study by Lee et al. [457] Eupafolin has been shown to inhibit the production of inflammatory mediators in a keratinocyte (HaCaT) cell line. Eupafolin has been shown to reverse the state of oxidative stress and the inflammatory response induced by PM. The activity of eupafolin in the treatment of skin inflammatory diseases in the molecular dimension is expressed in the form of suppression of intracellular ROS generation, NK-κB activation, cyclooxygenase-2 (COX-2) protein and gene expression, and prostaglandin E2 (PGE2) production in HaCaT cells. Unfortunately, eupafolin is poorly soluble in water, which impairs skin penetration. To address these issues, Lin et al. [458] synthesized a eupafolin nanoparticle delivery system (ENDS) and checked its activity using PM-treated mice. As nanoparticle carriers, a nontoxic polymer, Polyvinyl alcohol (PVA), and an acid-responsive cationic polymer, Eudragit E100, were used. The prepared ENDS exhibited improvement of antioxidant and anti-inflammatory properties in comparison to raw eupafolin.

Fullerene derivatives, in particular glycomodification of fullerenes, in addition to their antioxidant, antibacterial, and antiviral effects, have anti-inflammatory effects on PM-induced skin diseases [459]. The three water-soluble glycofullerenes with glucosides, galactosides, and mannosides sugar substituents with the hydrodynamic diameters of 69.3 ± 5.2, 103.2 ± 6.1, and 172.0 ± 17.7 nm, respectively, were synthesized and examined for their ability to attenuate PM-induced oxidative stress and inflammation in HaCaT keratinocytes. Glycofullerenes were synthethized using the Cu(I) alkyne–azide cycloaddition (CuAAC) method. PM-induced redox imbalance in keratocyte cells causes the activation of the mitogen-activated protein kinase (MAPK) and Akt (also known as protein kinase B) pathways followed by upregulation of the expressions of inflammatory proteins (ICAM-1, COX-2, HO-1, and PGE2, etc.). The PM-activated pathways were suppressed by pre-treatment with glycofullerenes. The results confirmed that the obtained glycofullerenes have a protective effect on the skin exposed to PM, thanks to their antioxidant and anti-inflammatory effects and the ability to maintain the expression of barrier proteins (filaggrin, involucrin, repetin, and loricrin), unlike unmodified fullerenes.

## 7. Impact of NPs on the Natural Environment

Taking into account the fact that the use of NMs in the cosmetics industry and pharmaceutical preparations is increasing every year, it is necessary to consider the possible threats that the generated nano-wastes have on the environment. Wastewater treatment plants that use microorganisms to remove organic waste are particularly at risk. It is known that some nanoparticles, i.e., Ag-NPs, have a strong bactericidal effect. Due to the possibility of NPs penetrating into groundwater, the assessment of NPs’ toxicity was performed mainly on aquatic organisms, e.g., zebrafish, *Chlorella* sp., *Catostomus commersonii*. 

TiO_2_-NPs, commonly added to sunscreens, have been observed to reduce the percentage of viable zebrafish embryos [460]. In the Iswarya et al. [461] study, the toxic impact of anatase and rutile NPs was investigated using freshwater microalgae, *Chlorella* sp. The authors noted the reduction in cell viability and chlorophyll content under the influence of NPs and UV radiation. SEM microscopic analysis revealed damage to the cell nucleus and cell membrane, as well as to chloroplasts and other internal organelles of the algae. ZnO-NPs [462] can alter heart function and induce a cellular stress response in gill tissue of the white sucker (*Catostomus commersonii*), a freshwater teleost fish. Exposure to ZnO-NPs resulted in an increase in ventilation index by ~30% and a decrease in cardiac acetylcholinesterase activity. The authors confirmed the cardio-respiratory toxicity of ZnO-NPs. The toxicity of Ag-NPs and TiO_2_-NPs against planktonic crustaceans, *Daphnia magna* [463], and the rainbow trout (*Oncorhynchus mykiss*) [464] was also assessed. It has been shown that exposure to NPs triggers oxidative stress mechanisms in internal organs. TiO_2_-NPs’ effects on eukaryotic cells were recently analyzed by Gojznikar et al. [465]. Carbon-based NPs are cytotoxic and can accumulate in mammalian organs, such as the lungs and kidneys. It was confirmed that NPs titanium, polystyrene, and fullerene induce oxidative stress [466]. The NPs, after absorption by plants and translocation, may enter the food chain and become biomagnified. On the other side, the synthesized amorphous iron oxide nanoparticles (AIONPs) from waste incense sticks ash (ISA) were useful for the remediation of Congo red dye from wastewater [467]. Using the adsorption method, more than 70% removal was achieved after one hour. The benefits and risks of using nanoparticles in agroindustry have been described in review articles [468,469,470].

## 8. Conclusions

In recent years, nanotechnology has made great progress in dermatology and cosmetology. This is a fairly new field, considering the fact that the Nanodermatological Society was established only in 2010 by an outstanding dermatologist, Dr. A. Nasir. NPs have been beneficial both as a standalone therapy and as a tool to enhance the effectiveness of pharmacological therapy. The skin is a large and accessible area of the body. The interaction of NPs with human skin and their possible penetration is being investigated from a toxicological perspective as well as a drug delivery route that reduces systemic side effects.

There are many benefits of using dermal and transdermal drug delivery systems enriched with NPs. Benefits include such properties as exceptional skin penetration ability and controlled drug release (depot effect) to skin and skin appendages. Owing to NPs, many benefits can be achieved, such as improvement in the solubilization capacity of active pharmaceuticals, increase in the bioavailability of practically insoluble drugs, drug protection against enzymatic and hydrolytic degradation, and enhancement of storage stability. Advantages for therapy involve an increase in skin penetration of many active, potent pharmaceuticals (hydrophilic/lipophilic), avoidance of hepatic first-pass metabolism and the gastrointestinal (GI) tract, the reduction in adverse effects (safer in hepato-compromised patients), and multiple dosing, on-demand drug delivery. Besides the above advantages, NPs in skin care products require special and expensive preparation techniques. Furthermore, we must take into account the lag time for the drug to penetrate the skin. Unfortunately, drugs that require high blood levels cannot be administered. As for disadvantages there are the following: disruption of the stratum corneum lipids’ integrity as the result of interfacial chemistry, limited epidermal targeting, the possibility of polymorphic as well as possible toxicity (cytotoxicity, phototoxicity, genotoxicity, carcinogenicity), the ability to accumulate in cells, and ROS generation and connected consequences. All these disadvantages are still unsolved challenges.

There are methods of detection and synthesis of NPs. Currently, the most interesting is so-called biogenic synthesis, which eliminates the use of toxic reagents and extends the pharmacological activity of NPs. To our knowledge, there are no preparations on the market that contain this type of NP. Single examples described in the paper come from innovative experimental works. There is no doubt that, considering green chemistry requirements, the biogenic synthesis of NPs belongs to future trends.

For transdermal delivery of active substances, healthy, undamaged skin provides a solid barrier against penetration. Therefore, dermatology and cosmetology are interested in the smallest possible size of NP that is able to use transcellular transport between corneocytes in the stratum corneum and transport through hair follicles. Hence, with the help of Langerhans cells, the way is opened to the lymph nodes. Of particular interest is the study of the observed adjuvant effect, immunosuppression, and the use of NPs as vaccine delivery systems. Taking into account contemporary needs, NPs should be further examined in the field of immunology.

In the treatment of deep wounds, not only is the antibacterial effect important, but also the need to supplement or rebuild cavities, e.g., after injuries, transplants, and deep wounds such as diabetic foot ulcers and bedsores. The analysis of the literature shows that the preparation of scaffolds designed specifically for the patient, e.g., with 3D printing, is still a challenge. However, it is not only about mechanical stability, but also about increasing angiogenesis, a key factor in wound healing. Commercial products containing BBG are most often used for this purpose. Therefore, further efforts are needed to explore other soft tissue repair options.

The use of nanomaterials in cosmetic and biomedical products has resulted in increased interest in the negative effects of this exposure on humans and the environment. Most attention is paid to the NPs of metal oxides, which are widely used as additives to photoprotective cosmetics. The release of metallic NPs into the environment can pose a serious threat to the ecosystem. The most extensively studied is the toxic effect of NPs on aquatic flora and fauna due to the possibility of their penetration into groundwater. So far, there are no data on the fate of nanoparticle carriers. Although it can be assumed that biodegradable and biocompatible lipid carriers do not pose a threat to the environment, easier access of plant and animal organisms to active substances carried by carriers may raise concerns.

## Figures and Tables

**Figure 1 ijms-23-15980-f001:**
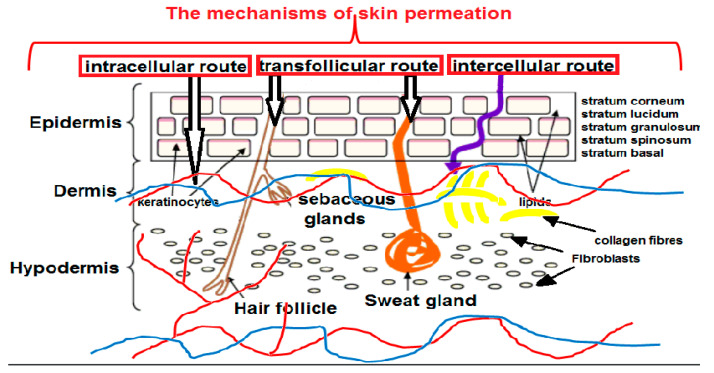
The schematic representation of the possible routes of the NPs permeation through the multilayered structure of the skin together with its main components and blood vessels (artery-red, vein-blue).

**Figure 2 ijms-23-15980-f002:**
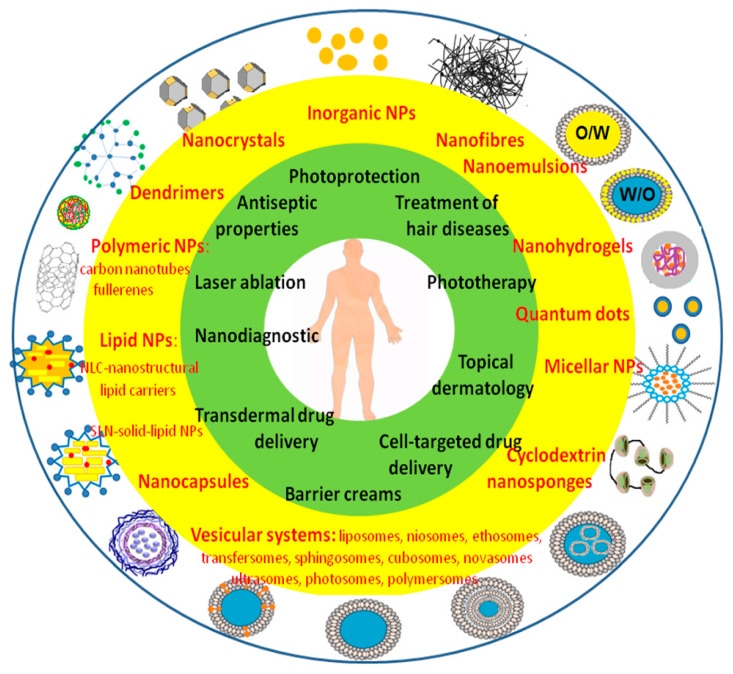
Schematic illustration of different nanoparticles and their applications for topical treatment in dermatology.

**Figure 3 ijms-23-15980-f003:**
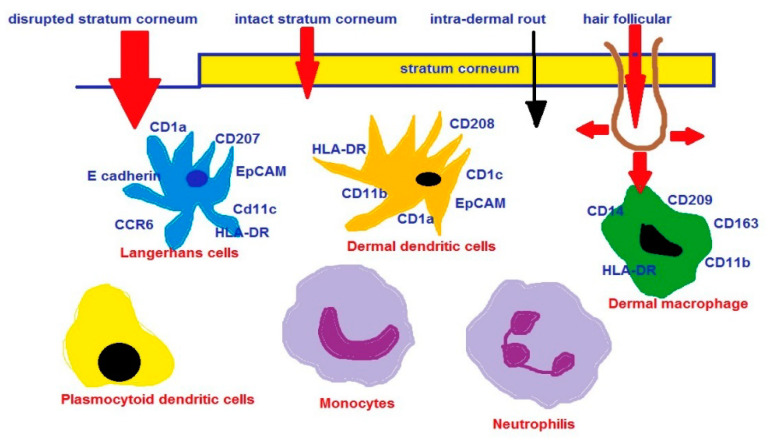
Skin penetration routes of NP-based vaccines together with subpopulations of APC (antigen-presenting cells).

**Figure 4 ijms-23-15980-f004:**
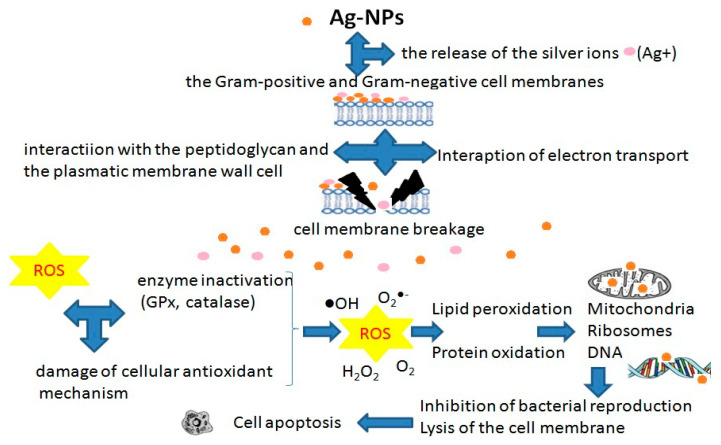
Mechanisms of Ag-NPs’ antimicrobial activity.

**Table 1 ijms-23-15980-t001:** Recent developments of drug-loaded NPs for topical treatments of skin diseases.

Nanocarriers	NPs Parameters	Drug Loaded	Skin Permeation Studies	Ref
**therapy of psoriasis**				
NLCs (Precirol VR ATO 5, LabrafacTM PG, TweenVR 20)	203.3 ± 1.20 nm; PDI 0.29 ± 0.04; EE% 100%	dithranol	IMQ induced a murine model of human psoriasis	[358]
liposomes (DOTAP + CHl)	111 ± 1.62 nm; PDI 0.27 ± 0.08; EE% 93 ± 2.12%; ZP 41.12 ± 3.56 mV	cyclosporine	IMQ induced a murine model of human psoriasis	[359]
1-liposomes (PC + CH), 2-niosomes (Span 80 + CH), 3-emulsomes (PC + CH + tristearin)	1-(368.5 ± 43 nm, PDI 0.136 ± 0.024, EE% 70.98 ± 2.36%); 2-(342.7 ± 35 nm, PDI 0.167 ± 0.045, EE% 54.30 ± 2.16%); 3-(172.8 ± 10 nm, PDI 0.116 ± 0.019; EE% 83.79 ± 3.58%)	capsaicin	hairless albino rat skin from the abdominal and dorsal area after removing underlying fat and subcutaneous tissues	[360]
liposomes (PC + OA), (PC + CH)	80–140 nm; PDI 0.14−0.37; ZP 6.64–2.56 mV; EE mol% 2.41–10.20%	methotrexate	the abdominal skin of newly born pig, the hairs, and lipid layer were removed	[361]
niosomes (Span 60 + CH)	477.8 nm; EE% 83.02%	diacerein	in vitro-the albino rats; the hairs, and subcutaneous tissues were removed	[362]
niosomes (Span 60 + CH)	369.73 ± 45.45 nm; EE% 90.32 ± 3.80%; ZP −36.33 ± 1.80 mV	acitretin	ex vivo- HaCaT cells (a keratinocyte cell line); in vivo-mouse tail model	[363]
niosome hydrogels (Span 20, Span 60, CH)	147.4 nm ± 5.6 nm; PDI 0.258 ± 0.02; ZP 48.9 mV ± 1.1 mV; Y% 90.42% ± 3.38%.	celastrol	female C57/BL6 mice, the dorsal hair was removed (a razor + depilatory cream)	[364]
ethosomes (soya lecithin + ethanol)	376.04 ± 3.47 nm; EE% 91.77 ± 0.02%	methotrexate + salicylic acid	ex-vivo-pig ear skin; in vivo-the shaved dorsal surface of Albino mice	[365]
ethosomes, liposomes	116 to 199 nm (liposomes); 146 to 381 nm (ethosomes); EE% ≥ 97.2% (liposomes), ≥77% (ethosomes)	anthralin (1,8-dihydroxy-9-anthrone)	in vitro-a dialysis membrane (MWCO: 6–8 kDa); ex vivo-Wistar male albino rats with abdominal skin freed from hair; the connective tissue, fat, and subcutaneous tissues	[366]
cationic liposomes (DC-CH, CH), anionic liposomes (egg lecithin, CH, tetramyristoyl cardiolipin)	100 nm; ZP + 25.8 mV, EE% 75.12% (cationic); ZP −28.5 mV, EE% 60.08% (anionic)	psoralen + UVA (PUVA)	IMQ-induced psoriatic plaque model	[367]
nanoemulsion	202.6 ± 11.59 nm; PDI 0.233 ± 0.01; EE% 76.57 ± 2.48%	methotrexate	in vivo, in vitro: male rabbits, male Sprague Dawley rats; the abdomen skin cleaned of lipids and connective tissues	[368]
nanoemugel	76.93 nm; PDI 0.121; ZP −20.5 mV	Curcumin + imiquimod	ex vivo, in vivo-BALB/c mice; the abdominal, and the dorsal skin hair shaved, the subcutaneous tissue, and fat at the dermis removed	[369]
nanoemusion	93.37 ± 2.58 nm; PDI 0.330 ± 0.025.	tacrolimus and kalonji oil	in vitro-a dialysis bag membrane with a molecular weight cut-off 12,000 Da; in vitro-A-431 cell lines; ex vivo-the dorsal portion of pig ear skin shaved; subcutaneous fat removed; in vivo-IMQ induced psoriasis model on BALB/c mice shaved on their back	[370]
the nanomiemgel composed of nanomicelle-NMI (with Vit.E TPGS) + nanoemulsion – NEM (olive oil + miglyol + Polysorbate80 + Transcutol)	229 ± 16 nm (NEM); 185 ± 10 nm (NMI); PDI 0.18 ± 0.06 (NEM); PDI 0.12 ± 0.08 (NMI); EE% 96.74 ± 4.24% (ACE-NEM); EE% 95.72 ± 3.58% (CAP-NEM); NMI, 94.88 ± 3.76% (ACE-NMI); 92.69 ± 3.08% (CAP-NMI)	aceclofenac (ACE) capsaicin (CAP)	ex vivo-human skin with a thickness of 0.5–0.1 mm; in vivo-C57BL/6 mice with shaved backs	[371]
liposomes (DDC642)	100 nm	RNA interference (RNAi) molecule	in vitro-psoriasis-induced keratinocytes, and melanocytes cultured	[372]
transfersomes	94.49 ± 6 nm–154.65 ± 8.46 nm; EE% 59.17 ± 5.03%	tacrolimus	ex vivo, in vivo-Albino Wistar Rat, the hairs trimmed	[373]
flexible liposomes (lecithin, Tween-80)	76.1 ± 0.5 nm; PDI 0.251 ± 0.009; EE% 99%	all-trans retinoic acid, betamethasone	in vitro-immortalized human keratinocytes (HaCaT), full-thickness skin from the ventral part of rats with removed extraneous subcutaneous fat; in vivo-BALB/c mice model induced by IQM	[374]
polymeric NPs (PLGA)	307.3 ± 8.5 nm; PDI 0.317; ZP −43.4 ± 2.6 mV; EE% 61.1 ± 1.9%; DL 1.9 ± 0.1%	apremilast	male Wistar albino rats	[375]
PLGA NPs	100 nm	indomethacin	rat skin/iontophoresis	[376]
polymeric micelles (poly(ethylene glycol)-*b*-oligo(desaminotyrosyl-tyrosine octyl ester suberate)-*b*-poly(ethylene glycol)	70 nm; PDI ≤ 0.22	paclitaxel	HaCaT (a cell line of human keratinocytes); human cadaver skin samples	[377]
polymeric micelles (methoxy-poly(ethylene glycol)-dihexyl substituted polylactide (MPEG-dihexPLA) diblock copolymer)	10–50 nm	tacrolimus	human skin	[378]
polymeric nanocapsules (Eudragit RS 100)	139 ±3.6 nm; ZP +11.38 ± 1.7 mV; EE% 81 ± 2%	dexamethasone	in vitro	[379]
Nanospheres; lipoglobules	70 nm	thymoquinone	in vitro-cell lines and IMQ-induced psoriatic plaque model	[380]
biogenically obtained Au-NPs (ethanolic extract of Woodfordia fruticosa flowers)	10–20 nm	myricetin quercetin ellagic acid	in vivo-Swiss albino mice, IMQ-induced psoriasis-like skin inflammation model; shaved dorsum surface of the mice skin	[381]
Au-NPs functionalized by 3-mercapto-1-propansulfonate (AuNPs-3MPS)	5 nm	methotrexate	in vivo-C57BL/6 mice (Charles River) with shaved backs; in vitro-acute toxicity studied on human skin equivalents (HSEs) after 21-day culture period (adult human keratinocytes seeded on a dermal substitute consisting of a collagen type I matrix and fibroblasts)	[382]
**therapy of vitiligo**				
liposomes (DC-CH, CH + sodium deoxy cholate)	120–130 nm; ZP +46.2 mV; EE% 74.09% (psoralen), 76.91% (resveratrol)	psoralen, resveratrol	B16F10 cell line	[383]
deformable liposomes	64.8 ± 1.3 nm; PDI 0.14 ± 0.03; ZP −27.0 ± 1.1; EE% 82.7 ± 0.4 (baicalin); 61.1 ± 1.4 nm; PDI 0.24 ± 0.01; ZP −38.1 ± 1.4; EE% 87.1 ± 1.2 (berberine)	baicalin, berberine	ex vivo-the epidermis (thickness ~200 μm) from newborn pig; in vitro-human immortalized keratinocytes (HaCaT)	[384]
elastic cationic niosomes (Tween61/CH/dodecyldimethyl ammonium bromide)	163.5 ± 1.8 nm; ZP (+) 37.0 ± 0.6 (PE); 101.5 ± 0.5 nm; ZP (+) 36.1 ± 7.1 (TPE)	human tyrosinase plasmid (PE) Tat/human tyrosinase plasmid (TPE)	the melanoma (B16F10) cell	[385]
microemulsion	18.26 nm	clobetasol propionate	in vivo-human	[386]
ethosomes	80 ± 1.2 nm; ZP −0.531 ± 0.10 mV; EE% 99.14 ± 0.32%	methoxsalen	ex vivo, in vivo-shaved skin of Wistar rats	[387]
**therapy of alopecia**				
chitosan nanoparticles	235.5 ± 99.9 nm; PDI of 0.31 ± 0.01; ZP +38.6 ± 6.0 mV	minoxidil sulphate	in vitro-porcine ear skin	[388]
NLCs	393.5 ± 36.0 nm; PDI < 0.4; ZP +22.5 ± 0.2 mV; EE% 86.9% (minoxidil), 99.9% (latanoprost)	minoxidil + latanoprost	in vitro-porcine ear skin	[389]
polymeric nanocapsules (poly-ε-caprolactone) /manual massage	197.8 ± 1.2 nm; PDI 0.15 ± 0.01; ZP −30.1 ± 1.8 mV; EE% 93.9 ± 0.4%	latanoprost	in vitro-the full-thickness skin of the porcine ears	[390]
NLCs (stearic acid + oleic acid)	281.4 ± 7.4 nm; PDI 0.207 ± 0.009; ZP −32.90 ± 1.23 mV; EE% 92.48 ± 0.31%; DL 13.85 ± 0.47%	minoxidil	in vitro-the abdominal full-thickness skins of male Sprague Dawley rats	[391]
polymeric NPs	90–300 nm	minoxidil	C57BL/6 mice	[392]
PLGA nanospheres	182–205 nm	hinokitiol, glycyrrhetinic acid, 6-benzyl-amino- purine	in-vitro, in-vivo -C3H mice, extracted human scalp skin	[393]
cationized gelatin microspheres	-	Tbx21 siRNA	in vivo-C3H/HeJ mice, a mouse model of alopecia areata; ex vivo-skin samples obtained from the peripheral region of the alopecic skin of patients	[394]
liposomes (saturated phospholipid + CH)	6.1 ± 1.8−16.6 ± 3.4 µm; PDE 88.6%	finasteride	in vitro-the full-thickness abdominal, and the dorsal mice skin after hair, and subcutaneous fat removal	[395]
**therapy of acne**				
Liposomes (egg PC + CH)	120 nm; ZP −43 mV	lauric acids (anti-*Propionibacterium acnes* (*P. acnes*)	in vitro, in vivo-a mouse ear model	[396]
SLNs	180 ± 2 nm; ZP 47 ± 4; EE% 100 ± 1%	retinoic acid	in vivo-a mutant strain of hairless mouse (rhino mice)	[397]
niosomes	-	benzoyl peroxide, clindamycin	in vivo-110 patients	[398]
**therapy of skin cancer**				
cationic lipid NPs (CTAB + stearic acid + monoolein)	196.90 ± 39.73 nm; PDI 0.26 ± 0.02; ZP +63.85 ± 12.37 mV; EE% 47.39 ± 2.52%	doxorubicin	in vitro-the pig ears skin with removed hair; in vivo-female nude BALB/c/iontophoresis	[399]
cationic liposomes/iontophoresis	192.6 ± 9.0 nm; PDI 0.326 ± 0.004; ZP 56.4 ± 8.0 mV; EE% (curcumin) 86.8 ± 6.0%	curcumin + STAT3 siRNA	in vitro-mouse melanoma cells (B16F10), ex vivo-porcine ear skin; in vivo- *C57BL/6 mice*	[400]
liposomes (lipoid S75 + oleic acid)	79.0 ± 4.1 nm; PID 0.12; ZP 40.06.7 mV; EE% 71.2 ± 10.9% (QUE); 72.1 ±6.6% (RSV)	quercetin (QUE)+ resveratrol (RSV)	in vitro-human dermal fibroblasts culture; in vivo-female CD-1 mice with the back skin shaved	[401]
Immunoliposome /iontophoresis	137 ± 25 nm; PDI 0.26 ± 0.04; ZP −6 mV	5-fluorouracil+ cetuximab	in vitro-A431 (EGFR positive) and B16F10 (EGFR negative) cell lines; in vitro- dermatomed skin of the outer portion of porcine ears; in vivo-immunosuppressed Swiss nude mice	[402]
Au-NPs	55.1 ± 5.1 nm; ZP −15 ± 1 mV	phytochemicals present in *Vitis vinifera* seeds	in vitro-A431 cancer cell line (skin carcinoma, human); HaCaT (normal, human immortalized keratinocyte cell line)	[403]
Fe_2_O_3_-NPs	10 ± 2.5 nm; ZP 7.9 ± 0.4 mV	epirubicin	in vitro-melanoma WM266cells; ex vivo-human cadaver skin without subcutaneous fat	[404]
polymeric NPs with EGFR antibody/photodynamic therapy	-	indocyanine green	in vitro-shaved the dorsal skin area of the Female CD1 mice	[405]
poly (lactic-co-glycolic acid) NPs	130.4 ± 10.5 nm; PDI 0.095; EE% 52 ± 3.34%	bromelain	in vivo-Male Swiss albino mice	[406]

Abbreviations: polydispersity index (PDI); soya phosphatidylcholine (PC); cholesterol (CH); the entrapment efficiency (EE%); N-(1-(2,3-dioleoyloxy) propyl)-N, N, N-trimethylammonium chloride (DOTAP); phosphatidylcholine (PC); oleic acid (OA); zeta potential (ZP); entrapment efficiency (%EE); drug loading (%DL); poly-(D,L-lactide-coglycolide) (PLGA); imiquimod (IMQ); trans-activating transcriptional (Tat,T); drug loading (DL); percent drug entrapment (PDE); cetyltrimethylammonium bromide (CTAB); conjugated epidermal growth factor receptor (EGFR).

**Table 2 ijms-23-15980-t002:** Types of commercial skin care nanoformulations with their active ingredients.

The Active Ingredients	Nanoformulations
vit.E, panthenol	nanocapsules
coenzyme Q10, vitamins (A, C and E), natural extracts (*Symphytum officinale*, *Camellia sinensis*, *Gingko biloba*, *Chondrus crispus*, *Rosa damascena*, *Aloe barbadensis*), natural oils (*Helianthus*, *Prunus Armeniaca*, *Symphytum officinale*), hyaluronic acid, SOD	liposomes
plant extracts (*Canadian Willow*, *Camellia sinensis*), olive oil, vit.E, hyaluronic acid, soy firming agent, UVA/UVB filters	nanospheres
gold powder (24 k), silk, plant extracts (*Coffee arabica*, *Aloe barbadensis*, *Cucurbita pepo*), vitamins (A, C and E), hyaluronic acid, plant extracts, plant stem cell extracts	Au-NPs
coenzyme Q10, natural oils (hemp, macadamia nut), plant extracts (*Leontopodium nivale*)	NLCs, SNLs
ZnO-NPs, Fe_2_O_3_-NPs, TiO_2_-NPs, plant extracts, vitamins	nanocomplexes
vitamins (E, B3, provitamin B5), UVA/UVB filters, bepanthol	nanoemulsions
plant extracts (*Gingko biloba*), oils (almond, lavender), natural compounds (caffeine, amino acids, polyphenols)	nanosomes

## Data Availability

Not applicable.

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
