# Peer review of "Nanoparticles for Topical Application in the Treatment of Skin Dysfunctions—An Overview of Dermo-Cosmetic and Dermatological Products"

_ijms, 2022, doi:10.3390/ijms232415980_

Round 1
Reviewer 1 Report
This review is on the use of nanoparticles in dermo-cosmetics and dermatological applications. It summarizes various therapies using nanoparticles to diagnose and treat skin diseases such as melanoma, acne, alopecia, vitiligo, and psoriasis and also anti-aging, UV-protectant cosmetics. The manuscript is well-intended and well-organized. A few suggestions are listed below:
1. Page 2-Line 71. “An example of the use of NPs in medicine is Fe2O3-NPs used as a contrast in magnetic resonance imaging” In addition to their use as MRI contrast agents they are also widely used for the magnetic hyperthermia and drug delivery vehicle in cancer treatment. Related information can be also added.
2. It is recommended to mention the use of bioactive glass nanoparticles for dermatological and soft tissue engineering applications (Subheading Inorganic NPs). In addition to silicate-based bioactive glasses especially borate-based bioactive glass nanoparticles have been widely used in wound healing applications due to their ability to enhance angiogenesis (approved by FDA). Due to their ability to release of therapeutic ions and drugs they find applications in soft tissue engineering applications.
3. Figure 1 and Figure 2 should be revised to improve the style and quality. It is also difficult to follow some of the labels in these drawings.
4. The manuscript summarizes the current state of understanding on the related topic. It is recommended to indicate existing gaps and future research directions.
Author Response
This review is on the use of nanoparticles in dermo-cosmetics and dermatological applications. It summarizes various therapies using nanoparticles to diagnose and treat skin diseases such as melanoma, acne, alopecia, vitiligo, and psoriasis and also anti-aging, UV-protectant cosmetics. The manuscript is well-intended and well-organized. A few suggestions are listed below:
- Page 2-Line 71. “An example of the use of NPs in medicine is Fe2O3-NPs used as a contrast in magnetic resonance imaging” In addition to their use as MRI contrast agents they are also widely used for the magnetic hyperthermia and drug delivery vehicle in cancer treatment. Related information can be also added.
Answer: Thank you very much for this suggestion. We agree with the reviewer that this aspect is very interesting and absolutely should be mentioned in our review. Actually, this can be subject for individual review. We added in introduction the following sentences:
“An example of the use of NPs in medicine is Fe2O3-NPs used as a contrast in magnetic resonance imaging (MRI) [22]. Fe2O3-NPs, similarly to other magnetic nanoparticles (MNPs), besides their use as MRI contrast agents, can be used as vehicles, combined with superconductors, in magnetic drug-delivery systems (MDDS). Due to the possibility of precision guiding MNPs by an external magnetic field to the required area, MDDS has become promising in cancer therapy. MNPs can not only effectively transport and delivered drugs with a high concentration in cancerous tissues but also generate heat through the oscillation of their magnetic pulse (44–47 °C) enabling the process of thermoablation of cancer cells (magnetic hyperthermia)[Flores-Rojas, G.G.; López-Saucedo, F.; Vera-Graziano, R.; Mendizabal, E.; Bucio, E. Magnetic Nanoparticles for Medical Applications: Updated Review. Macromol. 2022, 2, 374-390. doi:10.3390/macromol2030024]”.
- It is recommended to mention the use of bioactive glass nanoparticles for dermatological and soft tissue engineering applications (Subheading Inorganic NPs). In addition to silicate-based bioactive glasses especially borate-based bioactive glass nanoparticles have been widely used in wound healing applications due to their ability to enhance angiogenesis (approved by FDA). Due to their ability to release of therapeutic ions and drugs they find applications in soft tissue engineering applications.
Answer: Thank you very much for the above suggestion. This new issue enriched our review. We added the following:
Borate Bioactive Glasses (BBGs) are interesting biodegradable, bioactive materials for wound healing applications. Currently, BBGs are exploited for not only soft tissue (wound healing, nerve, and muscle regeneration) but also hard tissue (bone regeneration, drug delivery for osteomyelitis, or osteonecrosis treatment) engineering applications [Ege, D.; Zheng, K.; Boccaccini, A.R. Borate Bioactive Glasses (BBG): Bone Regeneration, Wound Healing Applications, and Future Directions. ACS Appl. Bio. Mater. 2022, 5, 3608-3622. doi: 10.1021/acsabm.2c00384]. In the case of BBGs, the wound healing process improves due to the ionic dissolution products of BBGs [Okonkwo, U.A.; DiPietro, L.A. Diabetes and Wound Angiogenesis. Int. J. Mol. Sci. 2017, 18, 1419. doi: 10.3390/ijms18071419], especially boron which is important in different stages of wound healing: stimulates angiogenesis [Schuhladen, K.; Stich, L.; Schmidt, J.; Steinkasserer, A.; Boccaccini, A. R.; Zinser, E. Cu, Zn Doped Borate Bioactive Glasses: Antibacterial Efficacy and Dose-Dependent in Vitro Modulation of Murine Dendritic Cells. Biomater. Sci. 2020, 8, 2143– 2155, doi: 10.1039/C9BM01691K], takes part in the synthesis of extracellular matrix, and stimulates the secretion of collagen and proteins; stimulates HUVEC proliferation and migration associated with the MAPK signal pathway [Hu, H.; Tang, Y.; Pang, L.; Lin, C.; Huang, W.; Wang, D.; Jia, W. Angiogenesis and Full-Thickness Wound Healing Efficiency of a Copper-Doped Borate Bioactive Glass/Poly(lactic- co-glycolic acid) Dressing Loaded with Vitamin E in Vivo and in Vitro. ACS Appl. Mater. Interfaces. 2018, 10, 22939-22950. doi: 10.1021/acsami.8b04903]; promotes keratinocyte migration [Mehrabi, T.; Mesgar, A.S.; Mohammadi, Z. Bioactive Glasses: A Promising Therapeutic Ion Release Strategy for Enhancing Wound Healing. ACS Biomater. Sci. Eng. 2020, 6, 5399-5430. doi: 10.1021/acsbiomaterials.0c00528]; upregulates of the vascular endothelial growth factor (VEGF) [Bi, L.; Rahaman, M.N.; Day, D.E.; Brown, Z.; Samujh, C.; Liu, X.; Mohammadkhah, A.; Dusevich, V.; Eick, J.D.; Bonewald, L.F. Effect of bioactive borate glass microstructure on bone regeneration, angiogenesis, and hydroxyapatite conversion in a rat calvarial defect model. Acta Biomater. 2013, 9, 8015-26. doi: 10.1016/j.actbio.2013.04.043]. Moreover, boron is also an antiseptic agent that aids the wound-healing process [Sengupta, S.; Michalek, M.; Liverani, L.; Švančárek, P.; Boccaccini, A. R.; Galusek, D. Preparation and Characterization of Sintered Bioactive Borate Glass Tape. Mater. Lett. 2021, 282, 128843– 128853, doi: 10.1016/j.matlet.2020.128843]. Mirragen microfibers made of boron 13-93B3 glass is an example of a commercial product approved by the U.S. Food and Drug Administration (FDA) in 2016.
BBGs can be also incorporated in polymeric scaffolds usually enriched with different dopants (copper, zinc, gallium, and silver), enhancing antimicrobial properties [Schuhladen, K.; Stich, L.; Schmidt, J.; Steinkasserer, A.; Boccaccini, A.R.; Zinser, E. Cu, Zn doped borate bioactive glasses: antibacterial efficacy and dose-dependent in vitro modulation of murine dendritic cells. Biomater. Sci. 2020, 8, 2143-2155. doi: 10.1039/c9bm01691k; Zhao, S.; Li, L.; Wang, H.; Zhang, Y.; Cheng, X.; Zhou, N.; Rahaman, M.N.; Liu, Z.; Huang, W.; Zhang, C. Wound dressings composed of copper-doped borate bioactive glass microfibers stimulate angiogenesis and heal full-thickness skin defects in a rodent model. Biomaterials 2015, 53, 379-91. doi: 10.1016/j.biomaterials.2015.02.112]. However, it should be remembered that this action is dose-dependent, and too low a level cannot inhibit bacterial growth [Rau, J.V.; De Bonis, A.; Curcio, M.; Schuhladen, K.; Barbaro, K.; De Bellis, G.; Teghil, R.; Boccaccini, A.R. Borate and Silicate Bioactive Glass Coatings Prepared by Nanosecond Pulsed Laser Deposition. Coatings 2020, 10, 1105. https://doi.org/10.3390/coatings10111105. ] whereas too high a dose could harm fibroblast and keratinocyte cells or reduce angiogenesis [Naseri, S.; Griffanti, G.; Lepry, W. C.; Maisuria, V. B.; Tufenkji, N.; Nazhat, S. N. Silver-Doped Sol-Gel Borate Glasses: Dose-Dependent Effect on Pseudomonas Aeruginosa Biofilms and Keratinocyte Function. J. Am. Ceram. Soc. 2022, 105, 1711– 1722. doi: 10.1111/jace.17802].
- Figure 1 and Figure 2 should be revised to improve the style and quality. It is also difficult to follow some of the labels in these drawings.
Answer: Figure 1 and 2 have been improved. I hope it is better now.
- The manuscript summarizes the current state of understanding on the related topic. It is recommended to indicate existing gaps and future research directions.
Answer: Thank you for this suggestion. In our conclusion, we noticed the necessity of further efforts toward the biogenic synthesis of NPs, offering production in an easy, ecological way, new still unknown, and maybe bioactive materials. A rather new direction is the use of NPs as Vaccine Delivery systems. Studying literature and collecting data for this review we notice a threat to the environment connected with the widespread of NPs applications, namely easier access of plant and animal organisms to active substances carried by even biodegradable and biocompatible NPs. We added also perspective concerning added information about inorganic borate bioactive glasses (BBGs) as follows: “In the treatment of deep wounds, not only the antibacterial effect is important, but also the need to supplement or rebuild cavities, e.g. after injuries, transplants and deep wounds, e.g. diabetic foot ulcers and bedsores. The analysis of the literature shows that the preparation of scaffolds designed specifically for the patient, e.g. with 3D printing is still a challenge. However, it is not only about mechanical stability, but also about increasing angiogenesis, a key factor in wound healing. Commercial products containing BBG are most often used for this purpose. Therefore, further efforts are needed to explore other soft tissue repair options.

Reviewer 2 Report
Comments:
According to the editor this review article can be published after minor revision.
1. The author please mention to the some biodegradable and non-biodegraded nanoparticles in an introduction part and line no is 74&75.
2. Line no 80 &81 include some relevant articles such as “ G. Gnanamoorthy, Virendra Kumar Yadav, Daoud Ali, Ezhumalai Parthiban, Gokhlesh Kumar, V. Narayanan, New orchestrated of X-CuTiAP (en, trien, ETA and DMA) nanospheres with enhanced photocatalytic and antimicrobial activities, Journal of Industrial and Engineering Chemistry (2022).; G. Gnanamoorthy, K. Ramar, Daoud Ali, Virendra Kumar Yadav, A. Jafar ahamed, Gokhlesh Kumar, Synthesis and effective performance of Photocatalytic and Antimicrobial activities of Bauhinia tomentosa Linn plants using of gold nanoparticles, Optical Materials, 123 (2022) 111945; Photocatalytic properties of amine functionalized Bi2Sn2O7/rGO nanocomposites, G. Gnanamoorthy, S. Muthamizh, K. Sureshbabu, S. Munusamy, A. Padmanaban, A. Kaaviya, R. Nagarajan, A. Stephen, V. Narayanan, Journal of Physics and Chemistry of Solids,118 (2018) 21-31.; Cytotoxicity, Removal of Congo Red Dye in Aqueous Solution Using Synthesized Amorphous Iron Oxide Nanoparticles from Incense Sticks Ash Waste, Virendra Kumar Yadav, G. Gnanamoorthy, Daoud Ali, Sweta Parimita Bera, Arpita Roy, Gokhlesh Kumar, Nisha Choudhary, Haresh Kalasariya, Anup Basnet (2022) Journal of Nanomaterials, 2022, 5949595, 12.
3. Once check grammar mistakes and typo and topo errors
4. What are benefits of the nanoparticles in the skin dysfunctions?
Author Response
According to the editor this review article can be published after minor revision.
- The author please mention to the some biodegradable and non-biodegraded nanoparticles in an introduction part and line no is 74&75.
Answer: In introduction part we added the following: In view of the growing trend of applying NMs in medicine, there is also an intensified interest in their toxic side effects, especially of those NPs that are not biodegradable, i.e. NPs of metals and metal oxides (in contrast to biodegradable NPs prepared from a variety of materials such as lipids, proteins, polysaccharides and synthetic biodegradable polymers such as starches, chitin/chitosan, or poly-(D,L-lactide-coglycolide).
- Line no 80 &81 include some relevant articles such as “ G. Gnanamoorthy, Virendra Kumar Yadav, Daoud Ali, Ezhumalai Parthiban, Gokhlesh Kumar, V. Narayanan, New orchestrated of X-CuTiAP (en, trien, ETA and DMA) nanospheres with enhanced photocatalytic and antimicrobial activities, Journal of Industrial and Engineering Chemistry (2022).; G. Gnanamoorthy, K. Ramar, Daoud Ali, Virendra Kumar Yadav, A. Jafar ahamed, Gokhlesh Kumar, Synthesis and effective performance of Photocatalytic and Antimicrobial activities of Bauhinia tomentosa Linn plants using of gold nanoparticles, Optical Materials, 123 (2022) 111945; Photocatalytic properties of amine functionalized Bi2Sn2O7/rGO nanocomposites, G. Gnanamoorthy, S. Muthamizh, K. Sureshbabu, S. Munusamy, A. Padmanaban, A. Kaaviya, R. Nagarajan, A. Stephen, V. Narayanan, Journal of Physics and Chemistry of Solids,118 (2018) 21-31.; Cytotoxicity, Removal of Congo Red Dye in Aqueous Solution Using Synthesized Amorphous Iron Oxide Nanoparticles from Incense Sticks Ash Waste, Virendra Kumar Yadav, G. Gnanamoorthy, Daoud Ali, Sweta Parimita Bera, Arpita Roy, Gokhlesh Kumar, Nisha Choudhary, Haresh Kalasariya, Anup Basnet (2022) Journal of Nanomaterials, 2022, 5949595, 12.
Answer: We added the following references:
4.2.2. Antimicrobials and wound healing
Recently, the photocatalytic and antibacterial properties have been described for new orchestrated of X-CuTiAP (X- ethylenediamine (en), triethylenetetramine (trien), ethanolamine (ETA) and dimethylamine (DMA)) nanospheres, and Au-NPs synthesized using Bauhinia tomentosa Linn extract by Gnanamoorthy et al. [Gnanamoorthy, G.; Kumar Yadav, V.; Ali, D.; Parthiban, E.; Kumar, G.; Narayanan, V. New orchestrated of X-CuTiAP (en, trien, ETA and DMA) nanospheres with enhanced photocatalytic and antimicrobial activities. J. Ind. Eng. Chem .2022, 110, 503-519. doi: 10.1016/j.jiec.2022.03.025; Gnanamoorthy, G.; Ramar, K.; Ali, D.; Kumar Yadav, V.; Jafar ahamed, A.; Kumar, G. Synthesis and effective performance of Photocatalytic and Antimicrobial activities of Bauhinia tomentosa Linn plants using of gold nanoparticles. Optical Materials,2022, 123, 111945. doi: 10.1016/j.optmat.2021.111945]. The authors studied the activity of the obtained NPs against Staphylococcus Aureus (St. a) (gram-positive), Pseudomonas aeruginosa (Ps. a) (gram-negative) pathogens, and Candida albicans (C. a) (fungal) pathogens; and ram-negative Escherichia coli (E. coli) and Gram-positive Staphylococcus aureus (S. aureus), respectively.
- Impact of NPs on the natural environment
On the other side, the synthesized amorphous iron oxide nanoparticles (AIONPs) from waste incense sticks ash (ISA) were useful for the remediation of Congo red dye from wastewater [465]. Using the adsorption method, more than 70% removal was achieved after one hour.
Kumar Yadav, V.; Gnanamoorthy, G.; Ali, D.; Parimita Bera,S.; Roy, A,; Kumar, G.; Choudhary, N.; Kalasariya, H.;Basnet, A. Cytotoxicity, Removal of Congo Red Dye in Aqueous Solution Using Synthesized Amorphous Iron Oxide Nanoparticles from Incense Sticks Ash Waste. J. Nanomater.2022, 2022, ID 5949595. doi: 10.1155/2022/5949595.
- Once check grammar mistakes and typo and topo errors
Answer: We did it.
- What are benefits of the nanoparticles in the skin dysfunctions?
There are many benefits that have been gathered for specific skin conditions and cosmetic applications. There are among others (in short), controlled drug release to skin and skin appendages (carrier systems), targeting of hair follicle-specific cell populations, and transcutaneous vaccination. All above is possible owing to surface functionalization (e.g., the binding to specific ligands). However, the most important is the improvement of skin penetration properties, depot effect with sustained drug release, and allowing specific cellular and subcellular targeting (transdermal gene therapy, and cancer therapy). Details we tried to describe in subchapters devoted to specific skin problems.

Reviewer 3 Report
This review provides a discussion on the application of various nanoparticles in dermatological and cosmetic products, in the therapy of different skin diseases, and in cosmetology as antiaging, and UV-protecting agents. Moreover, the Authors focused on the penetration of NPs through the skin (way and model of skin penetration and the influence of physical-chemical properties of NPs on skin penetration efficiency), on the application of NPs in medicine and as drug-loaded NPs for topical treatments of skin diseases. Finally, the Authors presented different types of nano-dermo-cosmetics.
In my opinion, this review is very interesting and will be of considerable interest to the readership of the International Journal of Molecular Sciences in the areas of nanoparticles for dermatological applications. The article was written very clearly and correctly. However, the Authors should develop and improve point 7. Impact of NPs on the natural environment. Additionally, I suggest a comparison of advantages and disadvantages of NPs in dermo-cosmetic and dermatological products. I think Conclusions should be more detailed.
I would like to recommend this article for publication in the International Journal of Molecular Sciences, but after minor revision.
Author Response
This review provides a discussion on the application of various nanoparticles in dermatological and cosmetic products, in the therapy of different skin diseases, and in cosmetology as antiaging, and UV-protecting agents. Moreover, the Authors focused on the penetration of NPs through the skin (way and model of skin penetration and the influence of physical-chemical properties of NPs on skin penetration efficiency), on the application of NPs in medicine and as drug-loaded NPs for topical treatments of skin diseases. Finally, the Authors presented different types of nano-dermo-cosmetics.
In my opinion, this review is very interesting and will be of considerable interest to the readership of the International Journal of Molecular Sciences in the areas of nanoparticles for dermatological applications. The article was written very clearly and correctly. However, the Authors should develop and improve point 7. Impact of NPs on the natural environment. Additionally, I suggest a comparison of advantages and disadvantages of NPs in dermo-cosmetic and dermatological products. I think Conclusions should be more detailed.
Answer: Thank you very much for all your suggestions. The authors appreciate the efforts the reviewer made in the aim to improve our manuscript. According to the reviewer's suggestions, we added advantages and disadvantages of NPs in skin care products and gave more details and future perspectives in the Conclusions. Considering chapter 7 is devoted to the “Impact of NPs on the natural environment”-We think that it is an interesting idea to develop this issue, however, the review become too long and at this moment includes 470 References. So, if the reviewer does not mind, we treat this suggestion as the subject of an individual paper.
I would like to recommend this article for publication in the International Journal of Molecular Sciences, but after minor revision.
Answer: Thank You very much for the recommendation and helpful suggestions.
